# Contrastive Order Learning: A General Framework for Ordinal Regression

Chaewon Lee [1]   BeomJun Shim [1]   Kwang Pyo Choi [2]   Chang-Su Kim [1]

## Abstract

We propose contrastive order learning (ConOrd), a contrastive learning framework for ordinal regression that integrates the strengths of contrastive learning and order learning. While contrastive learning effectively leverages all samples in a batch, it typically ignores the inherent ordering among rank labels. Conversely, order learning explicitly models label ordinality but often relies on local, margin-based comparisons, limiting its ability to capture global ordinal structure. ConOrd addresses these limitations by introducing a contrastive order loss with soft affinity and disparity weights based on rank differences, enabling fine-grained modeling of ordinal relationships across all sample pairs within a batch. Extensive experiments on a range of ordinal regression tasks, including facial age estimation, blind image quality assessment, and blind video quality assessment, demonstrate that ConOrd consistently achieves state-of-the-art performance and generalizes well across diverse ordinal regression scenarios. The source code is available at https://github.com/cwlee00/ConOrd.

## 1. Introduction

Ordinal regression is a task to estimate the discrete or continuous rank of an object instance. For example, facial age estimation aims to predict a person's age given their facial photograph, while image quality assessment predicts the quality score for an image. It is a fundamental problem frequently arising in many real-world scenarios, including facial age estimation (Moschoglou et al., 2017), health status scoring (Engemann et al., 2022), image and video quality assessment (Ying et al., 2021; Hosu et al., 2017), and gaze direction estimation (Wang et al., 2022).

Despite its wide applicability, ordinal regression poses in-

herent challenges: there is no clear distinction between adjacent ranks, and the semantic gap between neighboring labels can be subtle or ambiguous. It is hence difficult for a machine to learn discriminative representations that accurately reflect the underlying ordinal structure. To address these challenges, various methods have been proposed (Li & Lin, 2007; Rothe et al., 2018; Geng et al., 2013; Diaz & Marathe, 2019). Recently, order learning techniques (Lim et al., 2020; Lee & Kim, 2021; Lee et al., 2022) have achieved notable success. Among them, geometric order learning (GOL) (Lee et al., 2022) enforces metric and order constraints to arrange instances according to their ranks in an embedding space. However, as a margin-based pairwise approach, GOL generates no gradient once the margin is satisfied and cannot fully exploit the richer ordinal context available at the batch level, limiting its ability to capture fine-grained ordinal relationships.

Meanwhile, supervised contrastive learning (Khosla et al., 2020) has also been extended to ordinal regression, most notably in the RnC algorithm (Zha et al., 2023). While supervised contrastive learning relies on categorical labels to define positive and negative pairs, RnC constructs rank-aware pairs by comparing relative rank differences within a batch. Specifically, RnC selects an anchor and a positive sample, and treats samples whose rank differences from the anchor exceed that of the positive as negatives. However, this hard thresholding collapses ordinal distances into binary decisions, treating samples with both moderately and substantially larger rank gaps equally as negatives. As a result, RnC under-utilizes the full spectrum of ordinal information available within a batch.

In this paper, we propose contrastive order learning (ConOrd), which integrates the complementary strengths of order learning and contrastive learning while alleviating their respective limitations. Order learning explicitly models ordinal structure in an embedding space by encouraging attraction between samples with similar ranks and repulsion between dissimilar ones (Lee et al., 2022), but it typically relies on local supervision such as pairwise or margin-based losses, which can be insufficient for learning a globally consistent embedding. In contrast, contrastive learning effectively exploits batch-level relationships among samples, yet often ignores the ordered nature of labels and treats ordinal differences in a categorical manner. ConOrd bridges this gap by introducing a contrastive order loss that

[1]School of Electrical Engineering, Korea University, Seoul, Korea [2]Samsung Electronics, Seoul, Korea. Correspondence to: Chang-Su Kim <changsukim@korea.ac.kr>.

*Proceedings of the 43rd International Conference on Machine Learning*, Seoul, South Korea. PMLR 306, 2026. Copyright 2026 by the author(s).

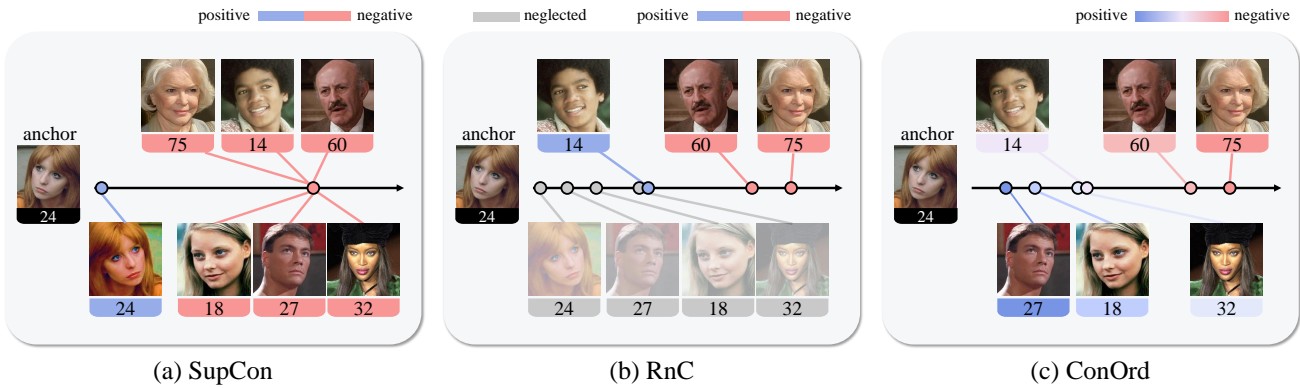

|  (a) SupCon  |  (b) RnC  |  (c) ConOrd  |

*Figure 1.* Comparison of three contrastive learning techniques: (a) supervised contrastive learning (SupCon) (Khosla et al., 2020), (b) Rank-N-Contrast (RnC) (Zha et al., 2023), and (c) the proposed ConOrd. SupCon considers the augmented view of an anchor as the only positive and treats all the others as negatives. RnC selects one sample as a positive and defines negatives as those with rank differences greater than the anchor–positive pair. Different from these existing techniques, ConOrd compares *all* samples in a batch by defining positive and negative samples in a *soft* manner.

assigns soft affinity and disparity weights based on rank differences, enabling all samples in a batch to contribute to learning with strengths modulated by their ordinal relationships. As illustrated in Figure 1, this design introduces soft, rank-aware interactions into a contrastive learning framework. By combining soft ordinal supervision with all-pairs contrastive comparison, ConOrd captures fine-grained ordinal structure while maintaining global consistency in the embedding space. Extensive experiments demonstrate that ConOrd achieves strong performance across a variety of ordinal regression tasks.

Our contributions are summarized as follows:

- We introduce contrastive order learning (ConOrd), a contrastive learning framework for ordinal regression that integrates ordinal structure into batch-wise representation learning.

- We propose a contrastive order loss that contrasts all sample pairs in a batch using soft affinity and disparity weights based on rank differences, enabling fine-grained ordinal supervision and the construction of a globally consistent embedding space.

- ConOrd achieves state-of-the-art performance on multiple ordinal regression benchmarks, including facial age estimation, blind image quality assessment (BIQA), and blind video quality assessment (BVQA), and demonstrates strong generalization across diverse regression tasks.

## 2. Related Work

### 2.1. Ordinal Regression

Ordinal regression aims to predict the ordinal labels or ranks of object instances, which have an inherent order. It has been used in various applications, including medical diag-

nosis (Wu et al., 2019; Liu et al., 2018), depth estimation (Fu et al., 2018), and facial age estimation (Rothe et al., 2018; Zhu et al., 2021). Early approaches either reformulated the problem as multiple binary classification tasks (Frank & Hall, 2001; Li & Lin, 2007) or dealt with it as a regression problem by adapting traditional classification loss functions (Rennie & Srebro, 2005; Rothe et al., 2018). To better exploit the ordinal properties of labels, subsequent methods have employed label distribution learning (Geng et al., 2013), mean-variance loss (Pan et al., 2018), soft ordinal labels (Diaz & Marathe, 2019), and probabilistic embeddings (Li et al., 2021).

However, many of these methods fail to capture inter-class relationships effectively, which may limit performance (Niu et al., 2016; Chen et al., 2017). To address this issue, distance-aware label embeddings (Shi et al., 2016), rank learning (Chen et al., 2017), and monotonic loss functions (Zhu et al., 2021) have been proposed. Also, Zhang et al. (2023) introduced an ordinal entropy regularizer, which promotes higher-entropy feature spaces while maintaining ordinal relationships. Meanwhile, evaluation metrics and loss designs that consider class proximity have been explored (Amigó et al., 2020), along with methods that tackle class imbalance (Nachmani et al., 2025). Through this progression, ordinal regression has come to be recognized — distinct from ordinary classification — as a predictive task that requires modeling both the order and the relative distances between classes.

### 2.2. Order Learning

Recently, order learning (Lim et al., 2020) has emerged as a promising approach to ordinal regression or rank estimation. In order learning, ordering relationships between object instances are learned, and the rank of an instance

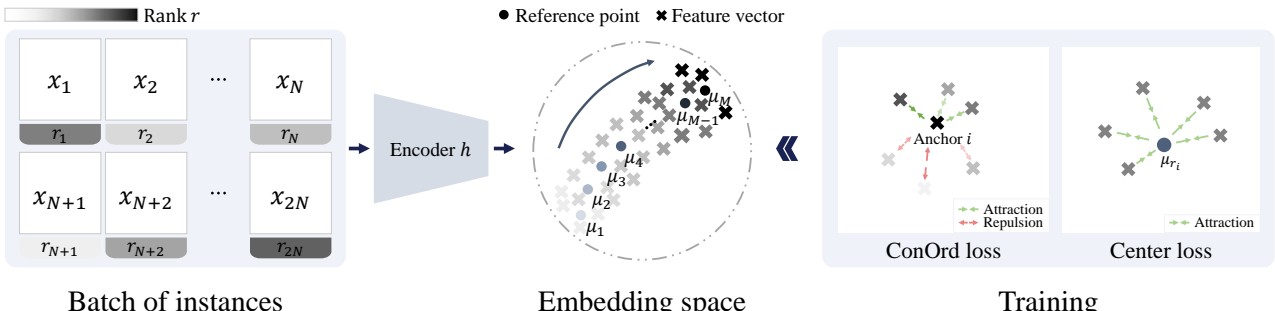

Figure 2. Overview of the proposed ConOrd framework for ordinal regression.

is estimated by comparing it with reference instances of known ranks. For reliable reference selection, Lee & Kim (2021) decomposed object embeddings into order and identity features and selected references with similar identity features. Shin et al. (2022) proposed a regression-based formulation to estimate a continuous relative rank between two references. Lee & Kim (2022) extended order learning to a weakly-supervised setting to cope with limited annotations, and Lee et al. (2024) proposed unsupervised order learning, which optimizes ordered clustering and embedding space construction alternately.

However, the direct comparison methods (Lim et al., 2020; Lee & Kim, 2021; Shin et al., 2022) should compare a test instance with multiple references, demanding considerable testing complexity, and do not consider metric relations between instances. To overcome these issues, Lee et al. (2022) proposed GOL that exploits metric, as well as order, relations to construct an embedding space and enables efficient rank estimation through a simple $k$-NN search in the embedding space.

### 2.3. Contrastive Learning

Contrastive learning aims to learn discriminative representations by modeling similarities and dissimilarities between object instances. It encourages the representations of similar (or positive) pairs to be pulled closer in an embedding space, while those of dissimilar (or negative) pairs to be pushed apart. Early methods (Chen et al., 2020; He et al., 2020) were largely explored in self-supervised scenarios, in which positive and negative pairs were constructed without requiring explicit labels.

To further leverage label information, supervised contrastive learning (Khosla et al., 2020) was introduced. Instead of solely relying on data augmentations to construct positive pairs, it also defines positive pairs from samples of the same class. The extension of contrastive learning to the fully supervised setting allows the learned feature space to reflect semantic structures more effectively, improving performance significantly in downstream tasks, such as image classification.

Although contrastive learning has achieved strong performance in various tasks, such as semantic segmentation (Liu et al., 2021), object detection (Xie et al., 2021), and medical imaging (Basak & Yin, 2023), its extension to regression and ordinal settings remains challenging. As contrastive objectives are primarily designed for categorical supervision, they often fail to capture the continuous ordering between samples. To address this, RnC (Zha et al., 2023) introduced rank-based pairings, but it relies only on relative ordering and ignores the magnitude of rank differences. Subsequent methods explored adaptive temperatures (Baek et al., 2024), data augmentation (Zheng et al., 2024), or multi-margin formulations (Pitawela et al., 2025), yet these approaches introduce additional hyperparameters or optimization complexity and may suffer from training instability. Building on these advances, we propose a soft-weighted contrastive scheme that explicitly incorporates rank differences to provide more stable and expressive ordinal supervision.

## 3. Proposed Algorithm

### 3.1. Problem Formulation and Overview

The objective of ordinal regression is to estimate the rank $r$ of a given instance $x$. Unlike classification, in ordinal regression, rank labels are naturally ordered and associated with inherent distance information (Lee et al., 2022). For example, consider three instances $x_i$, $x_j$, and $x_k$ with ranks $r_i = 10$, $r_j = 12$, and $r_k = 30$. In this case, there exists a natural ordering $r_i < r_j < r_k$, and the rank difference between $r_i$ and $r_j$ is smaller than that between $r_j$ and $r_k$, i.e., $|r_i - r_j| < |r_j - r_k|$. This observation highlights that effective learning for ordinal regression should account not only for relative ordering, but also for the magnitude of rank differences.

Motivated by this property, we aim to construct an embedding space in which instances are organized according to their ordinal relationships. To this end, we employ an encoder $h$ that maps each instance $x \in \mathcal{X}$ to a feature vector

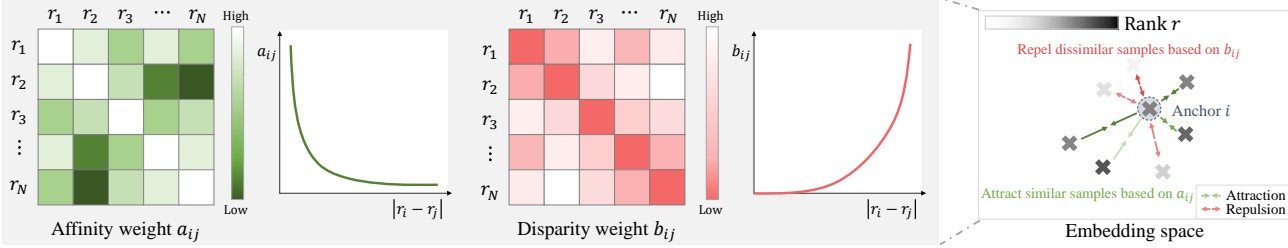

*Figure 3.* The ConOrd loss $L_{\text{ConOrd}}$ encourages the attraction of similar samples and the repulsion of dissimilar samples in the embedding space, by employing affinity weights $a_{ij}$ and disparity weights $b_{ij}$, respectively.

$z = h(x)$ in the embedding space, where features are $\ell_2$-normalized to lie on the unit sphere. Suppose that a training set $\mathcal{X}$ contains $M$ distinct ranks. We introduce $M$ learnable reference points $\{\mu_m\}_{m=1}^{M}$, where each $\mu_m$ represents a prototype associated with rank $m$. Building on this embedding space with reference points, we consider a contrastive formulation that can capture rank-dependent attraction and repulsion among instances, which guides the encoder to produce features that are both order-aware and well-structured. An overview of the proposed learning framework is illustrated in Figure 2.

We refer to a set of $2N$ training instances $\{x_i\}_{i=1}^{2N}$ as a batch. Unlike supervised contrastive learning (Khosla et al., 2020), which relies on data augmentation to define positive pairs, the proposed formulation operates directly on a batch of independently sampled instances and constructs the objective over all pairwise relationships without predefined positive or negative roles.

Let $I = \{1, \ldots, 2N\}$ be the index set of samples within a batch, and $z_i$ be the embedding vector of $x_i$. Also, $\kappa_{ij}$ denotes the similarity between $z_i$ and $z_j$ (*e.g.,* $\kappa_{ij} = z_i^T z_j$), and $\tau$ is a temperature parameter controlling the sharpness of the similarity. Then, the SupCon loss is given by

$$L_{\text{SupCon}} = -\frac{1}{2N} \sum_{i=1}^{2N} \frac{1}{|P(i)|} \sum_{p \in P(i)} \log \frac{e^{\kappa_{ip}/\tau}}{\sum_{j \in A(i)} e^{\kappa_{ij}/\tau}} \quad (1)$$

where $i$ is the index of an anchor, $A(i) = I - \{i\}$ is the set of the remaining indices, and $P(i)$ is the set of the indices of positive samples in the batch distinct from $i$.

In (1), the ratio $e^{\kappa_{ip}/\tau} / \sum_j e^{\kappa_{ij}/\tau}$ can be interpreted as the probability that the anchor $i$ is matched to the positive sample $p$ among all other samples in the embedding space. By minimizing the negative logarithm in (1), the matching probability is maximized. This suits the purpose of classification. On the other hand, in ordinal regression, it is desirable to match the anchor $i$ to another sample $j$ such that the rank estimation error $|r_i - r_j|$ is minimized. Thus, we can define the mean absolute error (MAE) loss for ordinal regression as

$$L_{\text{MAE}} = \frac{1}{2N} \sum_{i=1}^{2N} \frac{1}{|A(i)|} \log \frac{\sum_{j \in A(i)} |r_i - r_j| e^{\kappa_{ij}/\tau}}{\sum_{j \in A(i)} e^{\kappa_{ij}/\tau}}. \quad (2)$$

Note that $L_{\text{MAE}}$ involves the positive logarithm because it attempts to minimize the expected rank estimation error $\sum_j |r_i - r_j| \times (e^{\kappa_{ij}/\tau} / \sum_k e^{\kappa_{ik}/\tau})$.

## 3.2. Contrastive Order Learning

The naive loss $L_{\text{MAE}}$ in (2), however, makes the encoder training less reliable, for its positive logarithm tends to increase the magnitudes of gradients as the training goes on. To address this training issue while retaining its goal of matching the anchor to another sample of a similar rank in the embedding space, we design the ConOrd loss as

$$L_{\text{ConOrd}} = -\frac{1}{2N} \sum_{i=1}^{2N} \frac{1}{|A(i)|} \log \frac{\sum_{j \in A(i)} a_{ij} e^{\kappa_{ij}/\tau}}{\sum_{j \in A(i)} b_{ij} e^{\kappa_{ij}/\tau}}, \quad (3)$$

where $a_{ij}$ is an affinity weight, and $b_{ij}$ is a disparity weight.

This formulation is designed to satisfy three key requirements for ordinal contrastive learning: (i) attraction should decrease and repulsion should increase with the rank gap to preserve ordinal consistency (Lee et al., 2022), (ii) the weighting functions should vary smoothly with respect to the rank gap to promote stable optimization, and (iii) among multiple monotonic choices satisfying these requirements, a simple and robust instantiation is preferred in practice.

Following the above requirements, we instantiate the affinity and disparity weights as simple monotonic functions of the rank gap $|r_i - r_j|$. In particular, the affinity weight $a_{ij}$ is defined to decrease with the rank gap in order to promote similarity between samples of similar ranks, and is set as

$$a_{ij} = \frac{1}{(r_i - r_j)^2 + \epsilon}, \quad (4)$$

where $\epsilon$ is a small constant to prevent division by zero. Conversely, the disparity weight $b_{ij}$ is designed to increase with the rank gap to encourage separation between samples with distant ranks, and is instantiated as

$$b_{ij} = (r_i - r_j)^2. \quad (5)$$

We further set $\kappa_{ij} = -\|z_i - z_j\|_2^2$. Alternative choices of $\kappa_{ij}$, $a_{ij}$, and $b_{ij}$ are systematically evaluated in Appendix C.4.

The proposed ConOrd loss has the following properties.

*Table 1.* MAE comparison on age estimation datasets.

| ALGORITHM | MORPH II | CLAP2015 | AGEDB-DIR | UTK | CACD | ADIENCE |
|---|---|---|---|---|---|---|
| POE (LI ET AL., 2021) | 2.35 | 3.75 | 6.41 | 4.64 | 4.68 | 0.47 |
| PML (DENG ET AL., 2021) | 2.15 | 2.91 | 6.78 | 4.63 | 4.87 | 0.47 |
| MWR-G (SHIN ET AL., 2022) | 2.24 | 2.82 | 6.18 | 4.49 | 4.76 | 0.46 |
| GOL (LEE ET AL., 2022) | 2.17 | 3.38 | 6.21 | 4.35 | 4.52 | 0.43 |
| RANKSIM (GONG ET AL., 2022) | 3.10 | 5.55 | 6.51 | 4.93 | 4.68 | 0.60 |
| ORDINALCLIP (LI ET AL., 2022B) | 2.32 | 3.20 | 5.85 | 4.53 | 4.36 | 0.47 |
| ORDINALENTROPY (ZHANG ET AL., 2023) | 3.08 | 5.66 | 6.47 | 4.95 | 4.73 | 0.53 |
| RANKNCONTRAST (ZHA ET AL., 2023) | 2.47 | 4.72 | 6.14 | 4.74 | 5.14 | 0.40 |
| L2RCLIP (WANG ET AL., 2023) | 2.13 | 2.62 | - | - | - | 0.36 |
| NUMCLIP (DU ET AL., 2024) | 2.08 | 2.55 | 5.42 | 4.24 | **4.11** | 0.31 |
| CLOC (PITAWELA ET AL., 2025) | 2.84 | 4.14 | 6.87 | 4.81 | 4.66 | 0.41 |
| SLACE (NACHMANI ET AL., 2025) | 2.18 | 2.49 | 5.25 | 3.98 | 4.23 | **0.26** |
| CONORD (PROPOSED) | **1.96** | **2.46** | **5.15** | **3.92** | 4.18 | **0.26** |

*Table 2.* Performance comparison on five BIQA datasets.

| | BID | | CLIVE | | KONIQ10K | | SPAQ | | FLIVE | |
|---|---|---|---|---|---|---|---|---|---|---|
| ALGORITHM | SRCC | PCC | SRCC | PCC | SRCC | PCC | SRCC | PCC | SRCC | PCC |
| BMPRI (MIN ET AL., 2018) | 0.515 | 0.458 | 0.487 | 0.523 | 0.658 | 0.655 | 0.750 | 0.754 | 0.274 | 0.315 |
| SFA (LI ET AL., 2018) | 0.820 | 0.825 | 0.804 | 0.821 | 0.888 | 0.897 | 0.906 | 0.907 | 0.542 | 0.626 |
| DB-CNN (ZHANG ET AL., 2018) | 0.845 | 0.859 | 0.844 | 0.862 | 0.878 | 0.887 | 0.910 | 0.913 | 0.554 | 0.652 |
| PAQ-2-PIQ (YING ET AL., 2020) | - | - | 0.840 | 0.850 | 0.870 | 0.880 | - | - | 0.571 | 0.623 |
| HYPERIQA (SU ET AL., 2020) | 0.869 | 0.878 | 0.859 | 0.882 | 0.906 | 0.917 | 0.916 | 0.919 | 0.535 | 0.623 |
| UNIQUE (ZHANG ET AL., 2021) | 0.858 | 0.873 | 0.854 | 0.890 | 0.896 | 0.901 | - | - | - | - |
| MUSIQ (KE ET AL., 2021) | - | - | - | - | 0.905 | 0.919 | 0.917 | 0.920 | 0.640 | 0.721 |
| TRES (GOLESTANEH ET AL., 2022) | - | - | 0.846 | 0.877 | 0.915 | 0.928 | - | - | 0.554 | 0.625 |
| CONRTIQUE (MADHUSUDANA ET AL., 2022) | - | - | 0.845 | 0.857 | 0.894 | 0.906 | 0.914 | 0.919 | 0.580 | 0.641 |
| RE-IQA (SAHA ET AL., 2023) | - | - | 0.840 | 0.854 | 0.914 | 0.923 | 0.918 | 0.925 | 0.645 | 0.733 |
| QPT (ZHAO ET AL., 2023) | 0.842 | 0.852 | 0.857 | 0.881 | 0.912 | 0.927 | 0.916 | 0.921 | 0.551 | 0.635 |
| LQMAMBA-B (GUAN ET AL., 2024) | - | - | 0.837 | 0.875 | 0.895 | 0.913 | 0.912 | 0.914 | - | - |
| LODA (XU ET AL., 2024) | - | - | 0.876 | 0.899 | 0.932 | 0.944 | 0.925 | 0.928 | 0.578 | 0.679 |
| QCN (SHIN ET AL., 2024) | 0.892 | 0.890 | 0.875 | 0.893 | 0.934 | 0.945 | 0.923 | 0.928 | 0.644 | 0.741 |
| VISGA (SHI ET AL., 2025) | 0.904 | 0.923 | 0.882 | 0.912 | 0.931 | 0.940 | - | - | - | - |
| UNQA (CAO ET AL., 2025) | 0.881 | 0.914 | 0.874 | 0.903 | 0.893 | 0.915 | 0.913 | 0.917 | - | - |
| RICHIQA (MIN ET AL., 2025) | 0.900 | 0.909 | 0.894 | 0.912 | 0.938 | 0.950 | 0.923 | 0.929 | 0.583 | 0.684 |
| CONORD (PROPOSED) | **0.913** | **0.925** | **0.900** | **0.921** | **0.947** | **0.958** | **0.927** | **0.931** | **0.651** | **0.752** |

- **Rank-aware attraction and repulsion:** In Appendix A, the gradient of $L_{\text{ConOrd}}$ is derived as

$$\frac{\partial L_{\text{ConOrd}}}{\partial z_i} = \frac{1}{N\tau|A(i)|} \sum_{j \in A(i)} e^{\kappa_{ij}/\tau} \left( \frac{a_{ij}}{\alpha_i} - \frac{b_{ij}}{\beta_i} \right)(z_i - z_j) \tag{6}$$

where $\alpha_i = \sum_j a_{ij} e^{\kappa_{ij}/\tau}$ and $\beta_i = \sum_j b_{ij} e^{\kappa_{ij}/\tau}$. The factor $\left( \frac{a_{ij}}{\alpha_i} - \frac{b_{ij}}{\beta_i} \right)$ in (6) determines whether gradient descent moves $z_i$ toward or away from $z_j$. Since the affinity weight $a_{ij}$ is large for small rank gaps and the disparity weight $b_{ij}$ is large for large rank gaps, ConOrd attracts samples with similar ranks and repels samples with distant ranks, as illustrated in Figure 3. This provides a smooth ordinal interaction mechanism that generalizes the margin-based attraction and repulsion forces in GOL (Lee et al., 2022).

- **Locality induced by the exponential kernel:** The factor $e^{\kappa_{ij}/\tau}$ in (6) further modulates each pairwise contribution according to embedding proximity, because $\kappa_{ij} = -\|z_i - z_j\|_2^2$. Pairs that are already close in the embedding space receive stronger weights, whereas distant pairs are exponentially suppressed. Consequently,

ConOrd focuses its updates on the most informative ordinal relations — pairs that are close in rank and close in the embedding space — leading to fine-grained ordinal discrimination.

- **Contrasting all samples with soft weights:** A key strength of $L_{\text{ConOrd}}$ lies in its ability to contrast all samples in a batch through soft weighting, rather than relying on hard assignment of positives and negatives. Traditional contrastive losses (Chen et al., 2020; He et al., 2020; Khosla et al., 2020; Caron et al., 2020) select a positive sample and contrast it with all other samples in a batch, as in (1). In contrast, ConOrd assigns continuous affinity and disparity weights $a_{ij}$ and $b_{ij}$ to all sample pairs based on the rank difference $|r_i - r_j|$, allowing every pair $(i, j)$ to contribute to the loss with an influence modulated by ordinal similarity. By retaining all pairwise comparisons, ConOrd can exploit the full spectrum of ordinal relationships present within a batch.

- **Connection to RnC loss:** $L_{\text{ConOrd}}$ is conceptually related to the RnC loss (Zha et al., 2023), which also incorporates rank information into a contrastive framework. However,

*Table 3.* Comparison of BVQA results in the intra-dataset evaluation on LSVQ and in the cross-dataset evaluation on KonViD-1k, LIVE-VQC, CVD2014, and YouTube-UGC.

| ALGORITHM | LSVQ-TEST | | LSVQ-1080P | | KONVID-1K | | LIVE-VQC | | CVD2014 | | YOUTUBE-UGC | |
|---|---|---|---|---|---|---|---|---|---|---|---|---|
| | SRCC | PCC | SRCC | PCC | SRCC | PCC | SRCC | PCC | SRCC | PCC | SRCC | PCC |
| TVLQM (KORHONEN, 2019) | 0.772 | 0.774 | 0.589 | 0.616 | 0.732 | 0.724 | 0.670 | 0.691 | - | - | - | - |
| VSFA (LI ET AL., 2019) | 0.801 | 0.796 | 0.675 | 0.704 | 0.810 | 0.811 | 0.753 | 0.795 | 0.756 | 0.760 | 0.718 | 0.721 |
| VIDEVAL (TU ET AL., 2021) | 0.794 | 0.783 | 0.545 | 0.554 | 0.751 | 0.741 | 0.630 | 0.640 | - | - | - | - |
| PVQ (YING ET AL., 2021) | 0.827 | 0.828 | 0.711 | 0.739 | 0.791 | 0.795 | 0.770 | 0.807 | - | - | - | - |
| LI22 (LI ET AL., 2022A) | 0.852 | 0.854 | 0.772 | 0.788 | 0.843 | 0.835 | 0.793 | 0.811 | 0.817 | 0.811 | 0.802 | 0.792 |
| SIMPLEVQA (SUN ET AL., 2022) | 0.866 | 0.863 | 0.750 | 0.793 | 0.826 | 0.820 | 0.749 | 0.789 | 0.780 | 0.802 | 0.802 | 0.806 |
| FASTVQA (WU ET AL., 2022) | 0.876 | 0.877 | 0.779 | 0.814 | 0.859 | 0.855 | 0.823 | 0.844 | 0.805 | 0.814 | 0.730 | 0.747 |
| DOVER (WU ET AL., 2023) | 0.888 | 0.889 | 0.795 | 0.830 | 0.884 | 0.883 | 0.832 | 0.855 | 0.829 | 0.832 | - | - |
| KSVQE (LU ET AL., 2024) | 0.886 | 0.888 | 0.790 | 0.823 | - | - | - | - | - | - | - | - |
| MODULARBVQA (WEN ET AL., 2024) | 0.895 | 0.895 | 0.809 | 0.844 | 0.878 | 0.884 | 0.806 | 0.844 | 0.823 | 0.839 | 0.788 | 0.804 |
| CONORD (PROPOSED) | **0.904** | **0.904** | **0.818** | **0.851** | **0.889** | **0.892** | **0.836** | **0.865** | **0.839** | **0.845** | **0.805** | **0.828** |

similar to traditional contrastive losses, RnC relies on hard decisions to define positive and negative samples and employs binary weighting. The RnC loss is defined as

$$L_{\text{RnC}} = -\frac{1}{2N} \sum_{i=1}^{2N} \frac{1}{|A(i)|} \sum_{j \in A(i)} \log \frac{e^{\kappa_{ij}/\tau}}{\sum_{k \in \mathcal{N}(i,j)} e^{\kappa_{ik}/\tau}}, \quad (7)$$

where $\mathcal{N}(i,j) = \{k : |r_i - r_k| \geq |r_i - r_j|\}$. In RnC, each positive sample $j$ is contrasted against negatives $k$ satisfying $|r_i - r_k| \geq |r_i - r_j|$, which are aggregated equally in the denominator regardless of their rank gaps. As a result, RnC enforces relative ordering but does not distinguish how much farther negatives are from the anchor, limiting its ability to capture fine-grained ordinal structure. In contrast, ConOrd assigns continuous affinity and disparity weights $a_{ij}$ and $b_{ij}$ to all sample pairs, which can be viewed as a soft relaxation of the hard rank thresholds in RnC, enabling finer-grained ordinal supervision.

- **Choice of affinity and disparity weights:** The affinity and disparity weights depend only on the rank gap $d_{ij} = |r_i - r_j|$. In this work, we adopt quadratic forms, $a_{ij} = 1/(d_{ij}^2 + \epsilon)$ and $b_{ij} = d_{ij}^2$, which are simple, smooth, symmetric, and strictly monotonic functions of $d_{ij}$. This choice yields a balanced interaction between attraction and repulsion, assigning stronger attractive weights to small rank gaps while emphasizing repulsion for larger gaps. This design represents a stable instantiation within a broader class of monotonic weighting functions, and our experiments indicate that ConOrd is not sensitive to the specific functional form.

Additional properties and gradient analysis of $L_{\text{ConOrd}}$ are provided in Appendix A.

### 3.3. Training and Inference

In addition to $L_{\text{ConOrd}}$, we introduce the center loss (Nguyen et al., 2018) as a regularization term to further structure the embedding space. The center loss, which seeks to locate each reference point $\mu_m$ at the center of all instances with

rank $m$, is defined as

$$L_{\text{center}} = \sum_i \|z_i - \mu_{r_i}\|_2. \quad (8)$$

This encourages the encoder to produce a compact cluster in the embedding space for each rank.

Overall, we optimize the encoder parameters and reference points by minimizing the following total loss:

$$L_{\text{total}} = L_{\text{ConOrd}} + L_{\text{center}}. \quad (9)$$

In the inference phase, we estimate the rank of an unseen test instance based on the $k$-NN rule, as done in (Lee et al., 2022). We first extract the feature vector $z_t = h(x_t)$ of a test instance $x_t$. Then, in the embedding space, we find a set $\mathcal{N}$ of its $k$ nearest neighbors among all training instances in $\mathcal{X}$. Finally, the rank of $x_t$ is estimated by

$$\hat{r}_t = \frac{1}{k} \sum_{i:x_i \in \mathcal{N}} r_i. \quad (10)$$

## 4. Experimental Results

We apply ConOrd to three different ordinal regression tasks: facial age estimation, image quality assessment, and video quality assessment. Due to the space limitation, datasets and implementation details for each task are specified in Appendix B. Results on additional regression tasks, including temperature prediction and gaze direction estimation, as well as comparisons against deep imbalanced regression(DIR) methods on standard DIR benchmarks are also presented in Appendix C.

### 4.1. Facial Age Estimation

**Datasets:** We use six datasets of MORPH II (Ricanek & Tesafaye, 2006), CLAP2015 (Escalera et al., 2015), AgeDB-DIR (Moschoglou et al., 2017; Yang et al., 2021), UTK (Zhang et al., 2017c), CACD (Chen et al., 2015), and Adience (Levi & Hassner, 2015), as detailed in Appendix B.1.

**Comparison with state-of-the-art methods:** Recent state-of-the-art age estimation methods (Wang et al., 2023; Du et al., 2024) adopt a ViT-B backbone (Radford et al., 2021) due to its strong representational capacity. For fair comparison, ConOrd employs the same ViT-B encoder. As reported in Table 1, ConOrd achieves the best performance on all datasets except CACD under the MAE metric, demonstrating strong generalization across diverse age estimation benchmarks.

Notably, ConOrd outperforms OrdinalCLIP and Num-CLIP, which employ the same backbone but additionally leverage textual information to guide learning. It also achieves better or comparable performance compared with SLACE, an imbalance-aware ordinal regression loss that uses class-distribution information, under the same ViT-B backbone. Although the default ConOrd does not use explicit label-frequency statistics, it remains competitive with such imbalance-aware methods; Appendix C.4 further shows that a frequency-aware configuration of ConOrd can provide additional gains when such statistics are available.

## 4.2. Blind Image Quality Assessment

**Datasets:** We conduct BIQA experiments on five datasets of BID (Ciancio et al., 2010), CLIVE (Ghadiyaram & Bovik, 2015), KonIQ10k (Hosu et al., 2020), SPAQ (Fang et al., 2020), and FLIVE (Ying et al., 2020). The details are available in Appendix B.2.

**Comparison with state-of-the-art methods:** We use Spearman's rank order correlation coefficient (SRCC) and Pearson's linear correlation coefficient (PCC) to assess perceptual ranking and linearity. In Table 2, ConOrd outperforms the existing methods in terms of both metrics on all five datasets with no exception. Note that recent transformer-based techniques QPT (Zhao et al., 2023), LoDa (Xu et al., 2024), and QCN (Shin et al., 2024), as well as the Mamba-based method VISGA (Shi et al., 2025), the unified multi-modal quality assessment method UNQA (Cao et al., 2025), and the rich subjective information-based method RichIQA (Min et al., 2025) achieve high performance through large-scale pretraining, sophisticated network designs, or additional supervision. Nevertheless, the proposed algorithm consistently outperforms these techniques meaningfully, confirming the effectiveness of contrastive order learning.

## 4.3. Blind Video Quality Assessment

**Datasets:** We evaluate ConOrd on five widely used BVQA benchmarks — LSVQ (Ying et al., 2021), KoNViD-1k (Hosu et al., 2017), LIVE-VQC (Sinno & Bovik, 2018), CVD2014 (Nuutinen et al., 2016), and YouTube-UGC (Wang et al., 2019). For training, the LSVQ dataset is used. For evaluation, intra-dataset tests on LSVQ-test and

*Table 4.* Comparison of different contrastive losses on the CLAP2015 and LSVQ-1080p datasets under a controlled setting. All methods use the same ViT-B backbone, identical training protocol, and unified $k$-NN inference scheme.

| | CLAP2015 | LSVQ-1080P | |
| --- | --- | --- | --- |
| | MAE ($\downarrow$) | SRCC ($\uparrow$) | PCC ($\uparrow$) |
| $L_{\text{SupCon}}$ IN (1) (Khosla et al., 2020) | 2.625 | 0.614 | 0.682 |
| $L_{\text{OC}}$ (Baek et al., 2024) | 2.597 | 0.697 | 0.687 |
| $L_{\text{MMNP}} + L_{\text{CE}}$ (Pitawela et al., 2025) | 2.777 | 0.716 | 0.727 |
| $L_{\text{RNC}}$ IN (7) (Zha et al., 2023) | 2.531 | 0.812 | 0.843 |
| $L_{\text{ConOrd}} + L_{\text{CENTER}}$ IN (9) | **2.461** | **0.818** | **0.851** |

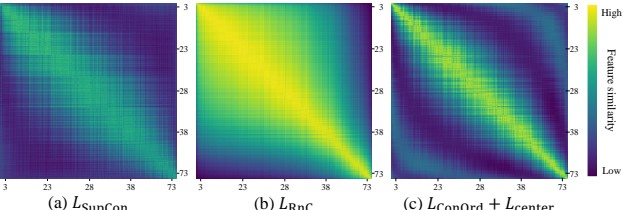

| (a) $L_{\text{SupCon}}$ | (b) $L_{\text{RnC}}$ | (c) $L_{\text{ConOrd}} + L_{\text{center}}$ |

*Figure 4.* Visualization of feature ordinality (Zha et al., 2023) on the CLAP2015 dataset.

LSVQ-1080p, as well as cross-dataset evaluations on the remaining four datasets, are conducted. More details on the datasets are in Appendix B.3.

**Comparison with state-of-the-art methods:** Table 3 presents a comprehensive comparison of ConOrd with state-of-the-art BVQA methods. Performance is measured using SRCC and PCC, as in BIQA. Again, ConOrd consistently outperforms all conventional methods across all datasets, achieving the best SRCC and PCC scores. While Modular-BVQA uses three distinct backbone networks of ResNet-18, SlowFast, and ViT-B, ConOrd relies on only two backbones of SlowFast and ViT-B. Despite the usage of fewer backbones, ConOrd outperforms ModularBVQA meaningfully. This indicates that ConOrd is capable of extracting more discriminative features with fewer computational resources, demonstrating its potential for practical deployment in real-world BVQA applications.

## 4.4. Ablations and Analyses

**Comparison of contrastive learning schemes:** In Table 4, we conduct a comparative analysis of both general-purpose (Khosla et al., 2020) and ordinal-regression-oriented (Zha et al., 2023; Baek et al., 2024; Pitawela et al., 2025) contrastive losses on the CLAP2015 and LSVQ-1080p datasets under a fully controlled setting. Specifically, all methods use the same ViT-B backbone, identical training protocol, and unified $k$-NN inference scheme, so that the performance differences can be attributed to the loss formulation rather than architectural or evaluation discrepancies.

$L_{\text{SupCon}}$ yields weak performance, as it disregards the ordinal property of rank labels. $L_{\text{OC}}$ shows slight improvements but remains limited, as it was primarily designed for medical

*Table 5.* Ablation study for the loss function in (9) on CLAP2015 and LSVQ-1080p.

| METHOD | $L_{\text{ConOrd}}$ | $L_{\text{center}}$ | CLAP2015 MAE ($\downarrow$) | LSVQ-1080P SRCC ($\uparrow$) | PCC ($\uparrow$) |
|---|---|---|---|---|---|
| I | ✓ | | 2.509 | 0.815 | 0.848 |
| II | | ✓ | 2.842 | 0.753 | 0.793 |
| III | ✓ | ✓ | 2.461 | 0.818 | 0.851 |

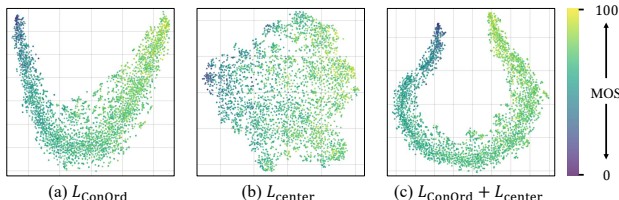

(a) $L_{\text{ConOrd}}$    (b) $L_{\text{center}}$    (c) $L_{\text{ConOrd}} + L_{\text{center}}$

*Figure 5.* t-SNE (Van der Maaten & Hinton, 2008) plots of the embedding spaces on LSVQ-1080p.

diagnostic datasets with relatively few ordinal levels. It does not scale effectively to tasks with many ordinal levels, *e.g.*, age estimation or BVQA. $L_{\text{MMNP}} + L_{\text{CE}}$ achieves moderate correlation gains; however, its performance is inconsistent across metrics due to the complexity of joint margin optimization. $L_{\text{RnC}}$ benefits from its ordinal-aware contrastive formulation and substantially outperforms these other existing losses. Finally, the proposed loss in (9) achieves the best results overall, showing that the all-pairs comparison through soft weighting helps learn better representations for ordinal regression.

These findings are further supported by the feature similarity matrices in Figure 4. As in Zha et al. (2023), the matrices are computed using the negative L2 norm between learned features on the CLAP2015 dataset. Representations are sorted by ground-truth ranks, so entries farther from the diagonal indicate larger rank differences. Compared to $L_{\text{SupCon}}$ and $L_{\text{RnC}}$, the proposed loss produces a clearer high-similarity band along the diagonal, confirming that the learned representations reflect the underlying ordinal structure more faithfully.

**Ablation study of loss function:** To assess the contribution of each component in (9), we conduct an ablation study in Table 5 and visualize the embedding space for each ablated method in Figure 5. $L_{\text{center}}$ alone fails to capture the ordinal nature of the task. While $L_{\text{ConOrd}}$ alone provides better results, its combination with $L_{\text{center}}$ further improves the results as the intra-class compactness is also considered. Both quantitatively and visually, the combined loss yields the most favorable results.

**Alternative choices for loss function:** We compare the behavior of the naive loss formulation $L_{\text{MAE}}$ in (2) with that of the proposed $L_{\text{ConOrd}}$ in (3). As shown in Figure 6, the positive logarithm in $L_{\text{MAE}}$ may make the training less

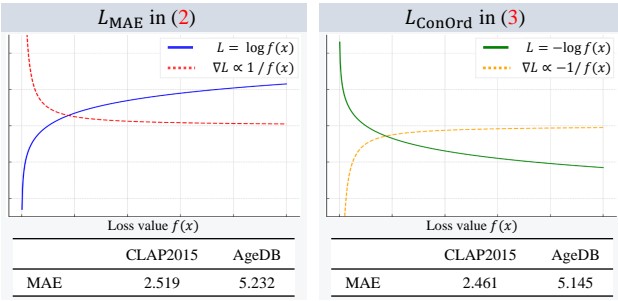

| $L_{\text{MAE}}$ in (2) | | $L_{\text{ConOrd}}$ in (3) | |
|---|---|---|---|
| $L = \log f(x)$ | | $L = -\log f(x)$ | |
| $\nabla L \propto 1/f(x)$ | | $\nabla L \propto -1/f(x)$ | |

| | CLAP2015 | AgeDB | | CLAP2015 | AgeDB |
|---|---|---|---|---|---|
| MAE | 2.519 | 5.232 | MAE | 2.461 | 5.145 |

*Figure 6.* Comparison of the losses in (2) and (3) on CLAP2015 and AgeDB-DIR.

*Table 6.* Comparison of alternative configurations of $L_{\text{ConOrd}}$ in (3) on CLAP2015.

| METHOD | $\kappa_{ij}$ | $a_{ij}$ | $b_{ij}$ | MAE ($\downarrow$) |
|---|---|---|---|---|
| I | $z_i^T z_j$ | $(\lvert r_i - r_j\rvert + \epsilon)^{-1}$ | $1$ | 2.516 |
| II | $z_i^T z_j$ | $(\lvert r_i - r_j\rvert + \epsilon)^{-1}$ | $\lvert r_i - r_j\rvert$ | 2.494 |
| III | $z_i^T z_j$ | $((r_i - r_j)^2 + \epsilon)^{-1}$ | $1$ | 2.483 |
| IV | $z_i^T z_j$ | $((r_i - r_j)^2 + \epsilon)^{-1}$ | $(r_i - r_j)^2$ | 2.518 |
| V | $-\lVert z_i - z_j\rVert_2^2$ | $(\lvert r_i - r_j\rvert + \epsilon)^{-1}$ | $1$ | 2.497 |
| VI | $-\lVert z_i - z_j\rVert_2^2$ | $(\lvert r_i - r_j\rvert + \epsilon)^{-1}$ | $\lvert r_i - r_j\rvert$ | 2.490 |
| VII | $-\lVert z_i - z_j\rVert_2^2$ | $((r_i - r_j)^2 + \epsilon)^{-1}$ | $1$ | 2.472 |
| VIII | $-\lVert z_i - z_j\rVert_2^2$ | $((r_i - r_j)^2 + \epsilon)^{-1}$ | $(r_i - r_j)^2$ | **2.461** |
| IX | $-\lVert z_i - z_j\rVert_2^2$ | $(\Delta^2 + \epsilon)^{-1}$ | $\Delta^2$ | 2.536 |
| X | $-\lVert z_i - z_j\rVert_2^2$ | LEARNABLE $a_{ij}$ | LEARNABLE $b_{ij}$ | 2.880 |

reliable, since the gradient magnitude may increase as the loss decreases. It means that the optimization process may become unstable over time, ultimately degrading the effectiveness of learning. Empirically, $L_{\text{ConOrd}}$ outperforms $L_{\text{MAE}}$ on CLAP2015 and AgeDB-DIR, confirming its improved stability.

Also, to investigate the impacts of different design choices in (3), we perform an ablation study on CLAP2015 by varying the components of $L_{\text{ConOrd}}$. The following observations can be made from the results in Table 6. First, negative squared Euclidean distance for $\kappa_{ij}$ consistently outperforms cosine similarity, suggesting that explicit distance-based measures better reflect ordinal relations in the embedding space. Second, method VIII, which incorporates squared differences into both $a_{ij}$ and $b_{ij}$, achieves the lowest MAE of 2.461. This indicates that emphasizing larger ordinal gaps during the contrastive optimization enhances model sensitivity to ordinal labels and improves regression accuracy. Method IX sets $\Delta$ to 0 if $r_i = r_j$, and to a positive threshold of 5 otherwise; these coarser approximations of ordinal differences are less effective than the proposed fine-grained modeling of rank gaps. Method X employs fully learnable $a_{ij}$ and $b_{ij}$ and yields poor results, suggesting that explicit encoding of ordinal structure is more effective than unconstrained learnable parameters. Appendix A provides a gradient-based justification for these configurations, while Appendix C.4 reports an extended table with additional configurations and results.

## 5. Conclusions

In this paper, we introduced contrastive order learning (ConOrd) as a general framework for ordinal regression that unifies order learning and contrastive learning within a single batch-wise formulation. By revisiting contrastive learning from an ordinal perspective, ConOrd incorporates rank-aware attraction and repulsion through soft affinity and disparity weights, enabling all sample pairs in a batch to contribute to learning in a principled and stable manner. We presented a simple yet effective instantiation of this framework. Through extensive experiments on diverse ordinal regression tasks, including facial age estimation, blind image quality assessment, and blind video quality assessment, we demonstrated that ConOrd consistently achieves strong performance and competitive generalization across benchmarks. We believe that contrastive order learning provides a flexible foundation for future advances in ordinal representation learning.

## Impact Statement

This work proposes a general method for ordinal regression that demonstrates strong performance across tasks, including facial age estimation, blind image quality assessment, and blind video quality assessment. However, caution is needed when deploying such models in sensitive domains. If used without appropriate safeguards, predictions involving human or facial attributes may raise ethical concerns, especially in the presence of biases within the training data. We recommend that future deployments incorporate fair evaluations and that the model be used as a decision-support tool rather than as an autonomous system.

## Acknowledgements

This work was supported by the National Research Foundation of Korea (NRF) funded by the Korea Government (MSIT) (No. RS-2024-00397293, RS-2022-NR068986), and by the AI Computing Infrastructure Enhancement (GPU Rental Support) User Support Program funded by MSIT (No. RQT-25-090187).

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

# A. Properties and Gradient Analysis of $L_{\text{ConOrd}}$ in (3)

We analyze the gradient of $L_{\text{ConOrd}}$ with respect to the embedding $z_i$. Using the chain rule, we have

$$\frac{\partial L_{\text{ConOrd}}}{\partial z_i} = \sum_{j \in A(i)} \frac{\partial L_{\text{ConOrd}}}{\partial \kappa_{ij}} \cdot \frac{\partial \kappa_{ij}}{\partial z_i}. \tag{11}$$

For simplicity, let $\alpha_i = \sum_{j \in A(i)} a_{ij} \exp(\kappa_{ij}/\tau)$ and $\beta_i = \sum_{j \in A(i)} b_{ij} \exp(\kappa_{ij}/\tau)$ in (3). Then,

$$L_{\text{ConOrd}} = -\frac{1}{2N} \sum_{i=1}^{2N} \frac{1}{|A(i)|} \log \frac{\alpha_i}{\beta_i}. \tag{12}$$

Computing the first term in the chain rule expression in (11),

$$\frac{\partial L_{\text{ConOrd}}}{\partial \kappa_{ij}} = -\frac{1}{2N|A(i)|} \left( \frac{1}{\alpha_i} \frac{\partial \alpha_i}{\partial \kappa_{ij}} - \frac{1}{\beta_i} \frac{\partial \beta_i}{\partial \kappa_{ij}} \right) \tag{13}$$

$$= -\frac{1}{2N|A(i)|} \left( \frac{1}{\alpha_i} \cdot \frac{a_{ij}}{\tau} \exp(\kappa_{ij}/\tau) - \frac{1}{\beta_i} \cdot \frac{b_{ij}}{\tau} \exp(\kappa_{ij}/\tau) \right) \tag{14}$$

$$= -\frac{1}{2N\tau|A(i)|} \exp(\kappa_{ij}/\tau) \left( \frac{a_{ij}}{\alpha_i} - \frac{b_{ij}}{\beta_i} \right). \tag{15}$$

Assuming that the similarity $\kappa_{ij}$ is defined as the negative squared Euclidean distance, *i.e.*, $\kappa_{ij} = -\|z_i - z_j\|_2^2$,

$$\frac{\partial \kappa_{ij}}{\partial z_i} = -2(z_i - z_j). \tag{16}$$

Thus, we have the gradient

$$\frac{\partial L_{\text{ConOrd}}}{\partial z_i} = \sum_{j \in A(i)} -\frac{1}{2N\tau|A(i)|} \exp(\kappa_{ij}/\tau) \left( \frac{a_{ij}}{\alpha_i} - \frac{b_{ij}}{\beta_i} \right) \cdot -2(z_i - z_j) \tag{17}$$

$$= \frac{1}{N\tau|A(i)|} \sum_{j \in A(i)} \exp(\kappa_{ij}/\tau) \left( \frac{a_{ij}}{\alpha_i} - \frac{b_{ij}}{\beta_i} \right) \cdot (z_i - z_j). \tag{18}$$

The following observations can be made from this gradient expression.

- **Attraction and repulsion induced by affinity and disparity weights:** The scalar factor $\left( \frac{a_{ij}}{\alpha_i} - \frac{b_{ij}}{\beta_i} \right)$ in (18) determines whether gradient descent pulls $z_i$ toward or pushes it away from $z_j$. Since the affinity weight $a_{ij}$ is large for small rank differences and the disparity weight $b_{ij}$ is large for large rank differences, pairs with similar ranks tend to yield positive factors (attraction), whereas pairs with distant ranks tend to yield negative factors (repulsion).

- **Locality induced by the exponential kernel:** The factor $\exp(\kappa_{ij}/\tau)$ in (18) modulates each pairwise contribution according to the current embedding distance, since $\kappa_{ij} = -\|z_i - z_j\|_2^2$. Pairs that are already close in the embedding space receive larger weights, while distant pairs are exponentially downweighted. Consequently, the gradient places greater emphasis on sample pairs that are both close in rank and close in the embedding space, enabling ConOrd to capture fine-grained ordinal distinctions more effectively.

- **Selective emphasis on informative pairwise relations:** Because strong forces arise only when two samples are close in the embedding space, the optimization primarily updates $z_i$ using pairs that the model already considers relevant. Combined with the affinity and disparity weights, this suppresses uninformative interactions with very distant ranks and allows ConOrd to adjust embeddings where ordinal information is most meaningful, without introducing instability as training progresses.

- **Choice of affinity and disparity weights:** The affinity and disparity weights depend only on the rank gap $d_{ij} = |r_i - r_j|$. We adopt the quadratic forms $a_{ij} = 1/(d_{ij}^2 + \epsilon)$ and $b_{ij} = d_{ij}^2$ because they are the simplest smooth, symmetric, and strictly monotonic functions of the ordinal distance. Quadratic growth provides a natural balance between near and far ranks — small gaps yield strong attractive weights, whereas large gaps contribute more to the repulsive term. While alternative monotonic choices (*e.g.,* linear or higher-order functions) are possible, we found that ConOrd is not highly sensitive to the exact form of these weights. Across the variants evaluated in Table 6, the quadratic form offers a good trade-off between stability and performance and performs consistently well across datasets.

This observation can be further understood from the gradient behavior of $L_{\text{ConOrd}}$. From (18), pair $(i, j)$ contributes attraction if $\frac{a_{ij}}{\alpha_i} > \frac{b_{ij}}{\beta_i}$, *i.e.,* $\frac{a(d_{ij})}{b(d_{ij})} > \frac{\alpha_i}{\beta_i}$, and repulsion otherwise. Since $\frac{\alpha_i}{\beta_i}$ is a shared threshold for anchor $i$, a key condition for the desired ordinal behavior is that $\frac{a(d)}{b(d)}$ decreases monotonically with $d$. This induces a single crossover rank gap, below which pairs are encouraged to attract and above which they are encouraged to repel, yielding a consistent ordinal interaction pattern for each anchor.

*Table 7.* Interpretation of different affinity and disparity weight configurations.

| Category | Methods | CLAP2015 MAE ($\downarrow$) | Monotone transition |
|---|---|---|---|
| Valid function class | I–VIII | 2.461–2.518 | ○ |
| Coarse approximation | IX | 2.536 | × |
| Unconstrained | X | 2.880 | × |

From this perspective, the configurations in Table 6 can be interpreted according to whether they satisfy this monotone transition property. As summarized in Table 7, weightings that satisfy it (Methods I–VIII) all perform comparably well, whereas configurations that break it (Method IX and Method X) show degraded performance. Thus, the residual variation within Methods I–VIII is better interpreted as a secondary effect of how each valid function emphasizes different rank gaps (and the choice of $\kappa_{ij}$), rather than evidence that the quadratic form is uniquely special.

# B. Experimental Details

## B.1. Facial Age Estimation

**Datasets:** Here, we provide more descriptions of each facial age estimation dataset.

- MORPH II (Ricanek & Tesafaye, 2006): As in Chang et al. (2011), we use 5,492 Caucasian images divided into training and test sets with a ratio of 8:2.

- CLAP2015 (Escalera et al., 2015): This dataset provides 4,691 facial images in total that are split into 2,476 for training, 1,136 for validation, and 1,079 for testing.

- AgeDB-DIR (Moschoglou et al., 2017; Yang et al., 2021): It contains 12.2K images for training, and the validation and test sets are balanced with 2.1K images. The age value ranges from 0 to 101.

- UTK (Zhang et al., 2017c): It consists of 20,000 facial images in a wide age range of [0,116]. We adopt the evaluation protocol in (Gustafsson et al., 2020; Berg et al., 2021).

- CACD (Chen et al., 2015): It provides 160k images of 2000 celebrities, which have an age range of [14, 62]. We use the training set specified in (Shin et al., 2022).

- Adience (Levi & Hassner, 2015): It has 26,580 facial images that are grouped into 8 ordinal classes: 0-2, 4-6, 8-13, 15-20, 25-32, 38-43, 48-53, and over 60-year-olds.

**Implementation details:** For age estimation, ViT-B in the CLIP algorithm (Radford et al., 2021) is employed as the encoder $h$. The model is trained using the Adam optimizer (Kingma & Ba, 2015) with a weight decay of 0.0005. We employ a cosine annealing scheduler (Huang et al., 2017) to adjust the learning rate. For data augmentation, only random horizontal flipping is applied.

## B.2. Blind Image Quality Assessment

**Datasets:** We evaluate the performance of the proposed ConOrd algorithm on the BIQA task using five datasets. For all datasets except FLIVE, we randomly split each dataset into train and test sets with a ratio of 4:1. Then, we repeat the training and evaluation over 10 different splits and report the median evaluation scores as done in previous methods (Zhao et al., 2023; Shin et al., 2024). For FLIVE, we employ the same evaluation protocol as in (Ying et al., 2020), where 30K images are used for training and 1.8K for testing.

- BID (Ciancio et al., 2010): It contains 586 images degraded by various types of realistic distortion (*e.g.*, motion blur, out-of-focus), with mean opinion scores (MOS) in the range $[0, 5]$.

- CLIVE (Ghadiyaram & Bovik, 2015): It consists of 1,169 natural images collected in diverse environments, annotated with MOS in the range [1, 100].

- KonIQ10k (Hosu et al., 2020): This dataset provides 10,073 images sampled from YFCC100M (Thomee et al., 2016), with MOS ranging from 1 to 100.

- SPAQ (Fang et al., 2020): Smartphone Photography Attribute and Quality (SPAQ) database includes 11,125 images taken with 66 different smartphones. Each image is assigned image attribute scores, but we use only the overall image quality scores in the range [0, 100].

- FLIVE (Ying et al., 2020): It is a large-scale BIQA dataset comprising approximately 40,000 images and 120,000 patches in the range [0, 100]. Following Ying et al. (2020), we only use the full-resolution images for training and testing, not the patches.

**Implementation details:** For BIQA, we also employ ViT-B from the CLIP algorithm (Radford et al., 2021) as the encoder $h$. The AdamW optimizer (Loshchilov & Hutter, 2017) is used with a weight decay of $2 \times 10^{-3}$. We use a cosine annealing scheduler (Loshchilov & Hutter, 2016) with a 5-epoch warm-up phase, during which the learning rate increases gradually to five times the initial value. For data augmentation, we use top-left, bottom-right, and center crops during training and use the average feature of the three cropped images.

## B.3. Blind Video Quality Assessment

**Datasets:** The following five BVQA datasets are used to train and evaluate the proposed algorithm.

- LSVQ (Ying et al., 2021): It is one of the largest datasets consisting of 39K videos, split into 28K for training and 11K for testing. The resolutions of videos are from 99p to 4K.

- KoNViD-1k (Hosu et al., 2017): It contains 1200 videos selected from YFCC-100M (Thomee et al., 2016) to cover various contents and distortions. All videos have a resolution of 540p.

- LIVE-VQC (Sinno & Bovik, 2018): It consists of 585 videos with various resolutions from 240p to 1080p.

- CVD2014 (Nuutinen et al., 2016): It is composed of 234 videos in five distinct scene types, filmed using 78 different cameras.

- YouTube-UGC (Wang et al., 2019): It provides about 1K video samples from user-generated contents on YouTube. The video resolutions range from 360p to 4K.

**Implementation details:** Recent state-of-the-art approaches in the BVQA task employ multiple backbone networks to capture diverse aspects of a video signal. Following this trend, we adopt a dual-backbone architecture in Figure 7. Specifically, we utilize ViT-B for the spatial feature extractor, which encodes $N$ sampled frames $I_i^1, I_i^2, ..., I_i^N$ into spatial feature vectors $z_i^{I_1}, z_i^{I_2}, ..., z_i^{I_N}$. These are then averaged to obtain a compact spatial representation $z_i^s$. For the temporal feature extractor, we adopt the Fast pathway network of the SlowFast video recognizer (Feichtenhofer et al., 2019) to extract temporal feature maps $z_i^{P_1}, z_i^{P_2}, ..., z_i^{P_M}$ from $M$ sampled clips $P_i^1, P_i^2, ..., P_i^M$. To enhance the temporal representations, we further refine the extracted temporal feature maps using a transformer module. This module performs inter-frame and intra-frame attention for each temporal feature map, yielding refined temporal features $\tilde{z}_i^{P_1}, \tilde{z}_i^{P_2}, ..., \tilde{z}_i^{P_M}$. The refined features are then averaged to obtain the final temporal representation $z_i^t$. Finally, the spatial and temporal representations are fused to generate the final feature vector $z_i$, which is used to compute the proposed ConOrd loss.

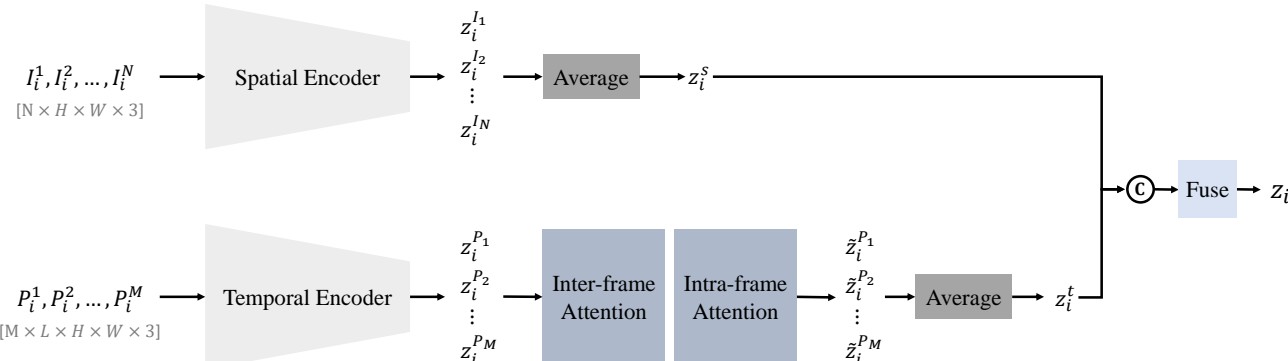

*Figure 7.* Network architecture for BVQA.

## B.4. Training and inference configurations

Table 8 summarizes the training and inference configurations for the proposed ConOrd algorithm.

*Table 8.* Training and inference configurations.

| Dataset | MORPH II | CLAP2015 | AgeDB-DIR | UTK | CACD | Adience | BID | CLIVE | KonIQ10k | SPAQ | FLIVE | LSVQ |
|---|---|---|---|---|---|---|---|---|---|---|---|---|
| Learning rate | $5 \times 10^{-6}$ | $5 \times 10^{-6}$ | $5 \times 10^{-6}$ | $5 \times 10^{-7}$ | $5 \times 10^{-7}$ | $2 \times 10^{-6}$ | $2 \times 10^{-6}$ | $2 \times 10^{-6}$ | $2 \times 10^{-6}$ | $2 \times 10^{-6}$ | $1 \times 10^{-6}$ | $5 \times 10^{-6}$ |
| Batch size | 128 | 128 | 128 | 128 | 128 | 128 | 32 | 32 | 64 | 64 | 64 | 16 |
| $\tau$ in (3) | 0.07 | 0.07 | 0.07 | 0.07 | 0.07 | 0.07 | 0.07 | 0.07 | 0.07 | 0.07 | 0.07 | 0.07 |
| $\epsilon$ in (3) | $10^{-7}$ | $10^{-7}$ | $10^{-7}$ | $10^{-7}$ | $10^{-7}$ | $10^{-7}$ | $10^{-7}$ | $10^{-7}$ | $10^{-7}$ | $10^{-7}$ | $10^{-7}$ | $10^{-7}$ |
| $k$ in (10) | 4 | 4 | 60 | 60 | 60 | 60 | 10 | 10 | 10 | 10 | 10 | 30 |

## C. More Experimental Results

### C.1. More Comparison on Additional Benchmarks

**Datasets:** We evaluate the performance of ConOrd on two additional regression datasets previously used in the evaluation of RnC (Zha et al., 2023).

- SkyFinder (Mihail et al., 2016; Chu et al., 2018): It is a dataset for predicting ambient temperatures from outdoor webcam images. It consists of 35,417 images taken by 44 different cameras under diverse weather and lighting conditions. The corresponding temperature values range from $-20\,°C$ to $49\,°C$. Following Zha et al. (2023), we split the dataset into 28,373 training and 3,522 test images.

- MPIIFaceGaze (Zhang et al., 2017a;b): It is a dataset for estimating gaze directions from face images, containing 213,659 face images collected from 15 participants during natural laptop use. We divide the dataset to construct a training set of 33,000 images and a test set of 6,000 images, ensuring no participant overlap across splits. Each image is annotated with a 2D gaze vector of pitch and yaw angles, where pitch angles range from $-40°$ to $10°$, and yaw from $-45°$ to $45°$.

For a fair comparison, we adopt the ResNet-18 backbone (He et al., 2016) as done in RnC (Zha et al., 2023). Table 9 shows that ConOrd achieves the best performance, outperforming all prior methods on both datasets.

*Table 9.* Performance comparison on the SkyFinder and MPIIFaceGaze datasets.

| Algorithm | SkyFinder MAE (↓) | MPIIFaceGaze Angular MAE (↓) |
|---|---|---|
| DEX (Rothe et al., 2015) | 3.58 | 5.72 |
| OR (Niu et al., 2016) | 2.92 | 5.86 |
| DLDL-v2 (Gao et al., 2018) | 2.99 | 5.47 |
| CORN (Shi et al., 2023) | 3.24 | 5.88 |
| RnC (Zha et al., 2023) | 2.86 | 5.27 |
| ConOrd (Proposed) | **2.65** | **4.98** |

**Regression examples**: We provide examples of regression results on the SkyFinder and MPIIFaceGaze datasets in Figure 8 and Figure 9, respectively.

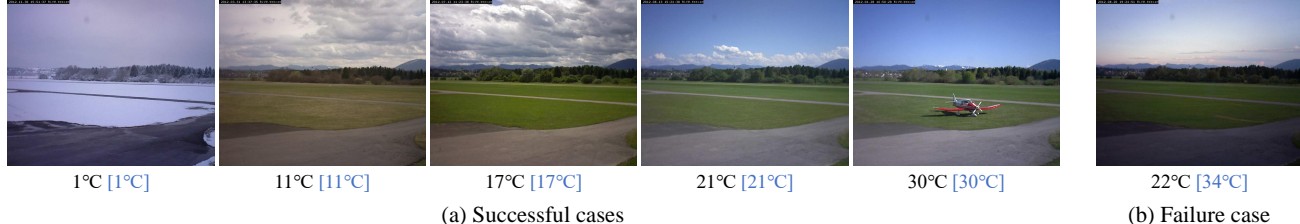

| 1°C [1°C] | 11°C [11°C] | 17°C [17°C] | 21°C [21°C] | 30°C [30°C] | 22°C [34°C] |
|---|---|---|---|---|---|

(a) Successful cases  (b) Failure case

*Figure 8.* Examples of regression results on the SkyFinder dataset. The estimated and ground-truth values are specified under each image: estimated [true].

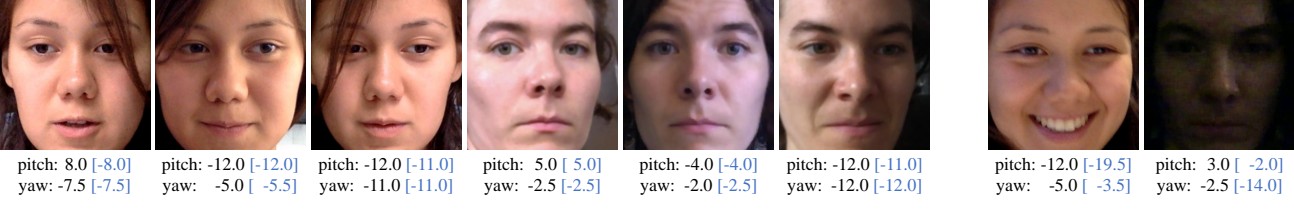

pitch: 8.0 [-8.0]  pitch: -12.0 [-12.0]  pitch: -12.0 [-11.0]  pitch: 5.0 [ 5.0]  pitch: -4.0 [-4.0]  pitch: -12.0 [-11.0]  pitch: -12.0 [-19.5]  pitch: 3.0 [ -2.0]
yaw: -7.5 [-7.5]  yaw: -5.0 [ -5.5]  yaw: -11.0 [-11.0]  yaw: -2.5 [-2.5]  yaw: -2.0 [-2.5]  yaw: -12.0 [-12.0]  yaw: -5.0 [ -3.5]  yaw: -2.5 [-14.0]

(a) Successful cases  (b) Failure case

*Figure 9.* Examples of regression results on the MPIIFaceGaze dataset. The estimated and ground-truth values are specified under each image: estimated [true].

## C.2. More Comparison with DIR Methods

To further examine the effectiveness of ConOrd, we compare it with representative deep imbalanced regression (DIR) methods on standard DIR benchmarks under controlled settings. This evaluation uses the same backbone and evaluation protocol for all competing methods within each benchmark, thereby isolating the effect of the learning objective.

**Datasets:** Following DIR methods, we evaluate ConOrd on IMDB-WIKI-DIR, STS-B-DIR, and NYUD2-DIR, in addition to AgeDB-DIR. These benchmarks cover age estimation, text similarity prediction, and depth estimation, while containing imbalanced training distributions.

- **IMDB-WIKI-DIR:** It is an age estimation benchmark derived from IMDB-WIKI (Rothe et al., 2018), consisting of face images with age labels. It contains 191,509 training images, 11,022 validation images, and 11,022 test images.

- **STS-B-DIR:** It is a natural language regression benchmark derived from the Semantic Textual Similarity Benchmark (STS-B) (Cer et al., 2017), where the task is to predict a continuous similarity score between sentence pairs. Following Yang et al. (2021), it contains 5,249 training pairs, 1,000 validation pairs, and 1,000 test pairs.

- **NYUD2-DIR:** It is a depth estimation benchmark constructed from the NYU Depth Dataset V2 (Silberman et al., 2012), which provides RGB images and depth maps for indoor scenes. Following standard practice (Hu et al., 2019), 50K images are used for training and 654 images are used for testing.

**Experimental protocol and results:** For AgeDB-DIR and IMDB-WIKI-DIR, we use ResNet-50 as the backbone. For STS-B-DIR, we use a BiLSTM with GloVe embeddings following Wang et al. (2018), and for NYUD2-DIR, we use a ResNet-50-based encoder–decoder (Hu et al., 2019). As shown in Table 10, ConOrd consistently outperforms competing DIR methods under identical conditions, confirming that the observed gains stem from the proposed objective.

*Table 10.* Comparison with ordinal and DIR methods on standard DIR benchmarks under matched settings.

| Method | AgeDB-DIR | IMDB-WIKI-DIR | STS-B-DIR | NYUD2-DIR |
|---|---|---|---|---|
| | MAE ($\downarrow$) | MAE ($\downarrow$) | MSE ($\downarrow$) | RMSE ($\downarrow$) |
| RankSim (Gong et al., 2022) | 7.02 | 7.50 | 0.903 | – |
| Ordinal Entropy (Zhang et al., 2023) | 7.46 | – | – | – |
| Rank-n-Contrast (Zha et al., 2023) | 6.14 | – | – | – |
| VIR (Wang & Wang, 2023) | 6.99 | 7.19 | 0.892 | 1.305 |
| ConR (Keramati et al., 2024) | 7.20 | 7.33 | – | 1.304 |
| SRL (Dong et al., 2025) | 7.22 | 7.69 | 0.877 | – |
| ConOrd | **6.02** | **7.06** | **0.868** | **1.294** |

**Comparison on BIQA and BVQA benchmarks:** We additionally compare ConOrd with DIR methods on BIQA and BVQA benchmarks. To ensure a fair comparison, all methods are implemented under the same backbone, training protocol, and evaluation pipeline as ConOrd. As shown in Table 11, ConOrd outperforms representative DIR methods on both BID and LSVQ-1080p. These results further confirm that the performance gains of ConOrd are attributable to the proposed objective rather than architectural or evaluation differences.

*Table 11.* Comparison with DIR methods on BID and LSVQ-1080p.

| Method | BID | | LSVQ-1080p | |
|---|---|---|---|---|
| | SRCC ($\uparrow$) | PCC ($\uparrow$) | SRCC ($\uparrow$) | PCC ($\uparrow$) |
| RankSim (Gong et al., 2022) | 0.892 | 0.909 | 0.814 | 0.848 |
| Ordinal Entropy (Zhang et al., 2023) | 0.886 | 0.909 | 0.809 | 0.845 |
| Rank-n-Contrast (Zha et al., 2023) | 0.892 | 0.906 | 0.812 | 0.843 |
| ConR (Keramati et al., 2024) | 0.885 | 0.909 | 0.812 | 0.846 |
| SRL (Dong et al., 2025) | 0.900 | 0.921 | 0.814 | 0.847 |
| ConOrd | **0.913** | **0.925** | **0.818** | **0.851** |

## C.3. Hyperparameter Analysis

**Performance according to temperature $\tau$ in (3):** Table 12 reports the performance of the proposed algorithm according to the temperature parameter $\tau$ in (3), which controls the smoothness of the representation distribution. We observe that the performance is relatively stable within a range $0.05 \leq \tau \leq 0.10$, achieving the best results when $\tau = 0.07$, with the minimal MAE on CLAP2015 and consistently high correlation scores on BID and LSVQ. With larger values (*e.g.,* $\tau \geq 1.0$), the performance degrades gradually across all datasets, indicating that excessively large temperature values reduce the effectiveness of the proposed contrastive learning objective.

*Table 12.* Performance of the proposed algorithm according to $\tau$.

|  | CLAP2015 (Age) | BID (BIQA) | | LSVQ-test (BVQA) | |
| --- | --- | --- | --- | --- | --- |
| $\tau$ | MAE ($\downarrow$) | SRCC ($\uparrow$) | PCC ($\uparrow$) | SRCC ($\uparrow$) | PCC ($\uparrow$) |
| 0.05 | 2.475 | 0.910 | 0.925 | 0.904 | 0.904 |
| 0.06 | 2.462 | 0.912 | 0.926 | 0.903 | 0.903 |
| 0.07 | 2.461 | 0.913 | 0.925 | 0.904 | 0.904 |
| 0.08 | 2.480 | 0.913 | 0.931 | 0.903 | 0.904 |
| 0.09 | 2.466 | 0.911 | 0.921 | 0.903 | 0.903 |
| 0.10 | 2.530 | 0.911 | 0.924 | 0.903 | 0.903 |
| 0.50 | 2.576 | 0.901 | 0.913 | 0.897 | 0.893 |
| 1.00 | 2.591 | 0.888 | 0.905 | 0.897 | 0.894 |
| 1.50 | 2.589 | 0.883 | 0.901 | 0.895 | 0.892 |
| 2.00 | 2.770 | 0.888 | 0.900 | 0.892 | 0.890 |

We further visualize how the performance varies with the temperature parameter $\tau$. As shown in Figure 10, both SRCC and PCC remain stable within a practical range ($0.05 \leq \tau \leq 0.10$) and begin to deteriorate only when $\tau$ becomes excessively large. The corresponding training curves also indicate consistent convergence behavior across all settings.

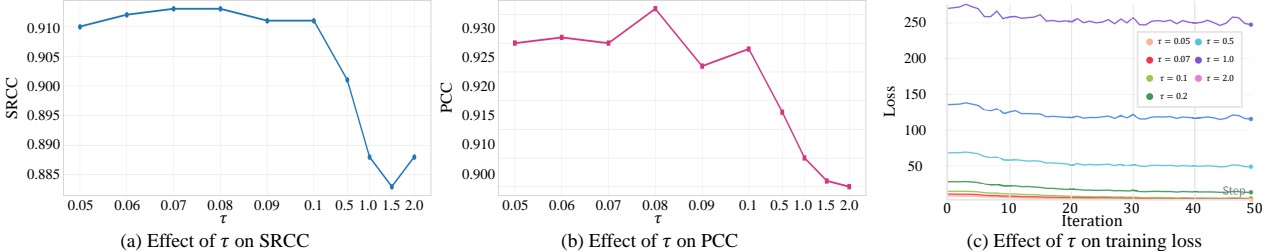

(a) Effect of $\tau$ on SRCC        (b) Effect of $\tau$ on PCC        (c) Effect of $\tau$ on training loss

*Figure 10.* Sensitivity of ConOrd to the temperature parameter $\tau$ on the BID dataset.

**Performance according to $\epsilon$ in (4):** Table 13 presents the results for varying values of $\epsilon$ in (4), which is used to prevent division by zero in the affinity weight computation. The model yields stable performance across a wide range of $\epsilon$ values, indicating that it is robust to the choice of $\epsilon$.

*Table 13.* Performance of the proposed algorithm according to $\epsilon$.

|  | CLAP2015 (Age) | BID (BIQA) | | LSVQ-test (BVQA) | |
| --- | --- | --- | --- | --- | --- |
| $\epsilon$ | MAE ($\downarrow$) | SRCC ($\uparrow$) | PCC ($\uparrow$) | SRCC ($\uparrow$) | PCC ($\uparrow$) |
| $10^{-3}$ | 2.494 | 0.915 | 0.928 | 0.904 | 0.904 |
| $10^{-4}$ | 2.483 | 0.925 | 0.926 | 0.904 | 0.904 |
| $10^{-5}$ | 2.485 | 0.914 | 0.929 | 0.903 | 0.903 |
| $10^{-6}$ | 2.472 | 0.914 | 0.930 | 0.902 | 0.903 |
| $10^{-7}$ | 2.461 | 0.921 | 0.925 | 0.904 | 0.904 |
| $10^{-8}$ | 2.484 | 0.910 | 0.928 | 0.903 | 0.903 |
| $10^{-9}$ | 2.483 | 0.911 | 0.925 | 0.904 | 0.904 |

To complement the quantitative results in Table 13, we additionally visualize the effect of $\epsilon$ in Figure 11, showing its

influence on BID performance and training behavior. SRCC and PCC remain nearly unchanged across a wide range of $\epsilon$, indicating strong robustness to this parameter. Training loss curves also show consistent convergence for all tested values.

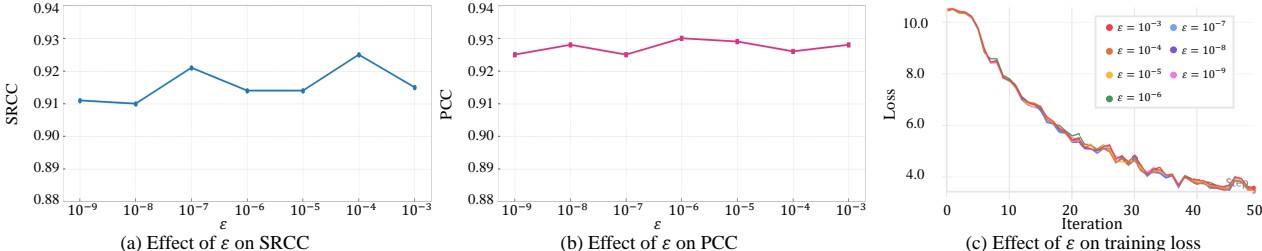

(a) Effect of $\varepsilon$ on SRCC    (b) Effect of $\varepsilon$ on PCC    (c) Effect of $\varepsilon$ on training loss

*Figure 11.* Sensitivity of ConOrd to $\epsilon$ on the BID dataset.

**Performance according to $k$ in (10):** Table 14 reports the results for varying values of $k$ used during $k$-NN inference. The performance remains stable across a broad range of $k$, with only marginal fluctuations across all datasets. The detailed configuration of the $k$ values is provided in Table 8.

*Table 14.* Performance across different values of $k$ used for $k$-NN inference.

| $k$ | CLAP2015 (Age) MAE ($\downarrow$) | AgeDB-DIR (Age) MAE ($\downarrow$) | BID (BIQA) SRCC ($\uparrow$) | PCC ($\uparrow$) | LSVQ-test (BVQA) SRCC ($\uparrow$) | PCC ($\uparrow$) |
|---|---|---|---|---|---|---|
| 4 | 2.461 | 5.232 | 0.910 | 0.924 | 0.894 | 0.892 |
| 10 | 2.483 | 5.159 | 0.913 | 0.925 | 0.901 | 0.900 |
| 20 | 2.496 | 5.158 | 0.909 | 0.923 | 0.903 | 0.903 |
| 30 | 2.503 | 5.158 | 0.910 | 0.924 | 0.904 | 0.904 |
| 40 | 2.496 | 5.154 | 0.910 | 0.923 | 0.904 | 0.904 |
| 50 | 2.487 | 5.158 | 0.910 | 0.923 | 0.904 | 0.904 |
| 60 | 2.486 | 5.145 | 0.910 | 0.923 | 0.904 | 0.904 |

## C.4. Additional analysis

**Performance according to loss balancing factors in (9):** Table 15 reports the impact of varying the weights of $L_{\text{center}}$ relative to $L_{\text{ConOrd}}$ across three benchmark datasets. The results indicate that the model is relatively robust to changes in the loss balancing factor, with only minor variations in MAE, SRCC, and PCC. Notably, setting both weights to 1.0 yields the best MAE on CLAP2015, suggesting that equal emphasis on the ordinal contrastive term and the intra-class compactness term provides the most effective trade-off for stable and accurate learning.

*Table 15.* Performance according to loss balancing factors.

| Loss combination | CLAP2015 (Age) MAE ($\downarrow$) | BID (BIQA) SRCC ($\uparrow$) | PCC ($\uparrow$) | LSVQ-test (BVQA) SRCC ($\uparrow$) | PCC ($\uparrow$) |
|---|---|---|---|---|---|
| $1.0 \times L_{\text{ConOrd}} + 0.5 \times L_{\text{center}}$ | 2.484 | 0.9127 | 0.9182 | 0.903 | 0.904 |
| $1.0 \times L_{\text{ConOrd}} + 0.8 \times L_{\text{center}}$ | 2.488 | 0.9023 | 0.9170 | 0.903 | 0.903 |
| $1.0 \times L_{\text{ConOrd}} + 1.2 \times L_{\text{center}}$ | 2.487 | 0.9005 | 0.9158 | 0.904 | 0.904 |
| $1.0 \times L_{\text{ConOrd}} + 1.5 \times L_{\text{center}}$ | 2.485 | 0.9036 | 0.9173 | 0.902 | 0.903 |
| $1.0 \times L_{\text{ConOrd}} + 1.0 \times L_{\text{center}}$ | 2.461 | 0.9025 | 0.9165 | 0.904 | 0.904 |

**Additional configurations for loss in (3):** Table 16 extends the analysis presented in Table 6 by including additional configurations of $\kappa_{ij}$, $a_{ij}$, and $b_{ij}$, as well as results on datasets beyond CLAP2015. These additional configurations explore alternative formulations, such as square-root and logarithmic scaling (methods XI and XII) and truncated weights (method XIII). The table provides a comprehensive view of how different design choices influence results. Overall, it reaffirms the effectiveness of the proposed configuration (method VIII) in accurately capturing ordinal relationships.

*Table 16.* Performance according to different configurations of $\kappa_{ij}, a_{ij}, b_{ij}$ in (3). Note that method IX sets $\Delta$ to 0 if $r_i = r_j$, and to a positive threshold of 5 otherwise.

| Method | $\kappa_{ij}$ | $a_{ij}$ | $b_{ij}$ | CLAP2015 (Age) MAE ($\downarrow$) | BID (BIQA) SRCC ($\uparrow$) | BID (BIQA) PCC ($\uparrow$) | LSVQ-test (BVQA) SRCC ($\uparrow$) | LSVQ-test (BVQA) PCC ($\uparrow$) |
|---|---|---|---|---|---|---|---|---|
| I | $z_i^T z_j$ | $\frac{1}{|r_i - r_j| + \epsilon}$ | $1$ | 2.516 | 0.9095 | 0.9263 | 0.902 | 0.901 |
| II | $z_i^T z_j$ | $\frac{1}{|r_i - r_j| + \epsilon}$ | $|r_i - r_j|$ | 2.494 | 0.9044 | 0.9180 | 0.904 | 0.904 |
| III | $z_i^T z_j$ | $\frac{1}{(r_i - r_j)^2 + \epsilon}$ | $1$ | 2.483 | 0.9078 | 0.9235 | 0.903 | 0.903 |
| IV | $z_i^T z_j$ | $\frac{1}{(r_i - r_j)^2 + \epsilon}$ | $(r_i - r_j)^2$ | 2.518 | 0.9088 | 0.9241 | 0.904 | 0.903 |
| V | $-\|z_i - z_j\|_2^2$ | $\frac{1}{|r_i - r_j| + \epsilon}$ | $1$ | 2.497 | 0.9086 | 0.9246 | 0.903 | 0.902 |
| VI | $-\|z_i - z_j\|_2^2$ | $\frac{1}{|r_i - r_j| + \epsilon}$ | $|r_i - r_j|$ | 2.490 | 0.9096 | 0.9257 | 0.903 | 0.902 |
| VII | $-\|z_i - z_j\|_2^2$ | $\frac{1}{(r_i - r_j)^2 + \epsilon}$ | $1$ | 2.472 | 0.9105 | 0.9242 | 0.902 | 0.902 |
| VIII | $-\|z_i - z_j\|_2^2$ | $\frac{1}{(r_i - r_j)^2 + \epsilon}$ | $(r_i - r_j)^2$ | 2.461 | 0.9128 | 0.9250 | 0.904 | 0.904 |
| IX | $-\|z_i - z_j\|_2^2$ | $(\Delta^2 + \epsilon)^{-1}$ | $\Delta^2$ | 2.536 | 0.6951 | 0.6948 | 0.834 | 0.830 |
| X | $-\|z_i - z_j\|_2^2$ | Learnable $a_{ij}$ | Learnable $b_{ij}$ | 2.880 | 0.6679 | 0.6854 | 0.882 | 0.881 |
| XI | $-\|z_i - z_j\|_2^2$ | $\frac{1}{\sqrt{|r_i - r_j|} + \epsilon}$ | $\sqrt{|r_i - r_j|}$ | 2.473 | 0.9099 | 0.9240 | 0.901 | 0.900 |
| XII | $-\|z_i - z_j\|_2^2$ | $\frac{1}{\log(1 + |r_i - r_j|) + \epsilon}$ | $\log(1 + |r_i - r_j|)$ | 2.500 | 0.9038 | 0.9199 | 0.901 | 0.900 |
| XIII | $-\|z_i - z_j\|_2^2$ | $\begin{cases} \frac{1}{|r_i - r_j| + \epsilon}, & \text{if } |r_i - r_j| < 10, \\ \frac{1}{10}, & \text{otherwise,} \end{cases}$ | $\begin{cases} |r_i - r_j|, & \text{if } |r_i - r_j| < 10, \\ 10, & \text{otherwise.} \end{cases}$ | 2.500 | 0.8958 | 0.9143 | 0.900 | 0.899 |

**Effect of $L_{\text{center}}$:** We evaluate the impact of the center loss by applying $L_{\text{center}}$ to $L_{\text{SupCon}}$, $L_{\text{RnC}}$, and $L_{\text{ConOrd}}$. As shown in Table 17, ConOrd performs strongly even without the center term, confirming that its gains mainly arise from the loss design itself. Adding $L_{\text{center}}$ yields small improvements and does not alter the relative ranking among methods. Overall, the center loss acts as a mild stabilizer rather than a key performance factor.

*Table 17.* Ablation of the effect of the center loss $L_{\text{center}}$.

| Method | CLAP2015 (Age) MAE ($\downarrow$) | BID (BIQA) SRCC ($\uparrow$) | BID (BIQA) PCC ($\uparrow$) | LSVQ-1080p (BVQA) SRCC ($\uparrow$) | LSVQ-1080p (BVQA) PCC ($\uparrow$) |
|---|---|---|---|---|---|
| $L_{\text{SupCon}}$ | 2.625 | 0.819 | 0.876 | 0.614 | 0.682 |
| $L_{\text{SupCon}} + L_{\text{center}}$ | 2.610 | 0.815 | 0.875 | 0.622 | 0.689 |
| $L_{\text{RnC}}$ | 2.531 | 0.892 | 0.906 | 0.812 | 0.843 |
| $L_{\text{RnC}} + L_{\text{center}}$ | 2.745 | 0.892 | 0.906 | 0.808 | 0.839 |
| $L_{\text{ConOrd}}$ | 2.509 | 0.909 | 0.925 | 0.815 | 0.848 |
| $L_{\text{ConOrd}} + L_{\text{center}}$ | 2.461 | 0.913 | 0.925 | 0.818 | 0.851 |

**Initialization of reference points $\mu_m$:** We assess several initialization strategies for $\mu_m$. On the BID dataset, random, zero, truncated normal, and Kaiming normal initializations yield nearly identical results, indicating that the method is largely insensitive to initialization.

*Table 18.* Performance across different initialization schemes for $\mu_m$ on BID.

| Init. method | Random | Zeros | Trunc. Normal | Kaiming Normal |
|---|---|---|---|---|
| SRCC | 0.913 | 0.910 | 0.913 | 0.910 |
| PCC | 0.925 | 0.922 | 0.925 | 0.925 |

We further test initializing $\mu_m$ with the per-rank mean feature. Although this variant offers a small performance gain, it incurs a substantial overhead due to the extra dataset pass. Thus, random initialization is a more efficient choice.

*Table 19.* Random initialization versus mean-feature initialization.

| Method | SRCC | PCC | Time (s) |
|---|---|---|---|
| Random init. | 0.913 | 0.925 | $1.21 \times 10^{-3}$ |
| Mean-feature init. | 0.915 | 0.932 | 5.25 |

**Resilience to reduced training data:** To assess robustness under limited ordering information, we progressively subsample the AgeDB-DIR training set and compare the resulting MAE performance — evaluated on the full test set — of ConOrd with SupCon and RnC. As shown in Table 20, ConOrd consistently outperforms both baselines across all sampling ratios. The advantage is most pronounced in low-data settings (*e.g.,* ratios of 0.1 and 0.3), suggesting that ConOrd learns more sample-efficient and stable ordinal representations when supervision is scarce.

*Table 20.* MAE results on AgeDB under different training-set sampling ratios, where each ratio denotes the proportion of training data used.

| Sampling ratio | $L_{\text{SupCon}}$ | $L_{\text{RnC}}$ | $L_{\text{ConOrd}} + L_{\text{center}}$ |
|---|---|---|---|
| 0.1 | 6.948 | 6.953 | 6.690 |
| 0.3 | 6.022 | 6.009 | 5.881 |
| 0.5 | 5.855 | 5.655 | 5.574 |
| 0.8 | 5.457 | 5.287 | 5.261 |
| 1.0 | 5.361 | 5.192 | 5.145 |

**Robustness to data corruption:** We adopt the corruption process defined in the ImageNet-C protocol (Hendrycks & Dietterich, 2019) and apply it to the AgeDB-DIR test set to evaluate robustness under data degradation. All methods are trained on the clean AgeDB-DIR training data, and MAE is measured on corrupted versions of the test images across 19 corruption types and severity levels 0–5. As shown in Table 21, ConOrd achieves the best performance under clean conditions (severity 0) and exhibits the slowest degradation as severity increases. Even at the highest corruption level (severity 5), ConOrd maintains a lower MAE than SupCon and RnC, indicating stronger robustness to corrupted inputs.

*Table 21.* MAE results on AgeDB-DIR under test-time data corruptions.

| Corruption severity level | $L_{\text{SupCon}}$ | $L_{\text{RnC}}$ | $L_{\text{ConOrd}} + L_{\text{center}}$ |
|---|---|---|---|
| 0 | 5.361 | 5.192 | 5.145 |
| 1 | 6.454 | 6.253 | 5.993 |
| 2 | 7.273 | 7.127 | 6.718 |
| 3 | 8.196 | 7.957 | 7.430 |
| 4 | 9.648 | 9.311 | 8.589 |
| 5 | 11.577 | 11.165 | 10.187 |

While Tables 20 and 21 summarize the numerical results, the corresponding visualizations in Figure 12 help illustrate the relative performance trends. In both reduced-data and corruption scenarios, ConOrd shows a consistently favorable margin over the baselines.

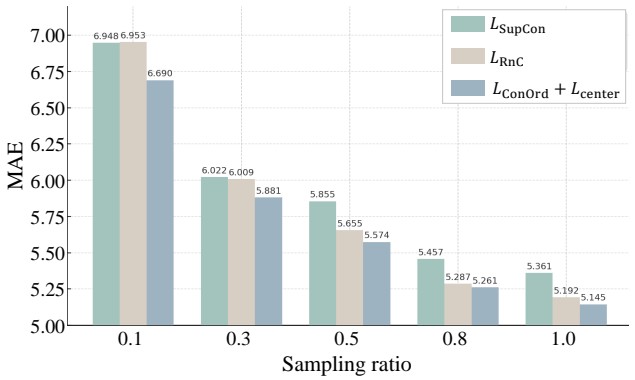
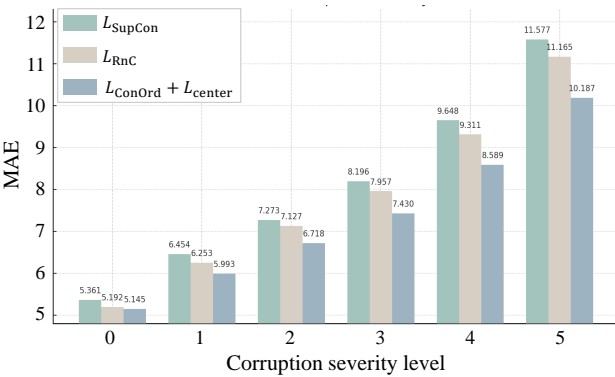

(a) Resilience to reduced training data.

(b) Robustness to data corruption.

*Figure 12.* Comparison of SupCon, RnC, and ConOrd under reduced supervision and test-time corruptions.

**Standard deviation of performance:** To assess the reliability of ConOrd, we report the mean and standard deviation across multiple random seeds. We use five seeds for the age estimation (CLAP2015) and BVQA (LSVQ-test) benchmarks, and ten seeds for the BIQA (BID) task. Table 22 summarizes the resulting variability. ConOrd exhibits stable performance across datasets.

*Table 22.* Mean and standard deviation of ConOrd across multiple random seeds.

| | CLAP2015 (Age) | BID (BIQA) | | LSVQ-test (BVQA) | |
|---|---|---|---|---|---|
| | MAE ($\downarrow$) | SRCC ($\uparrow$) | PCC ($\uparrow$) | SRCC ($\uparrow$) | PCC ($\uparrow$) |
| $L_{\text{SupCon}}$ | $2.6058 \pm 0.0121$ | $0.8182 \pm 0.0052$ | $0.8754 \pm 0.0035$ | $0.7576 \pm 0.0034$ | $0.7660 \pm 0.0032$ |
| $L_{\text{RnC}}$ | $2.5324 \pm 0.0202$ | $0.8901 \pm 0.0260$ | $0.9041 \pm 0.0237$ | $0.9024 \pm 0.0006$ | $0.9004 \pm 0.0011$ |
| ConOrd | $2.4698 \pm 0.0122$ | $0.9118 \pm 0.0243$ | $0.9255 \pm 0.0183$ | $0.9040 \pm 0.0007$ | $0.9036 \pm 0.0009$ |

**t-SNE visualization of learned embeddings:** Figure 13 provides a qualitative t-SNE comparison of embeddings learned by $L_{\text{SupCon}}$, $L_{\text{RnC}}$, and $L_{\text{ConOrd}} + L_{\text{center}}$ on the BID dataset. While t-SNE does not preserve global geometry and should be interpreted with caution, some overall trends can be observed. SupCon yields largely mixed points with no apparent ordering, and RnC shows a coarse progression but with some overlap between neighboring quality levels. ConOrd produces a more coherent progression of points along a smooth trajectory, suggesting that its embedding space reflects ordinal structure more clearly under this visualization.

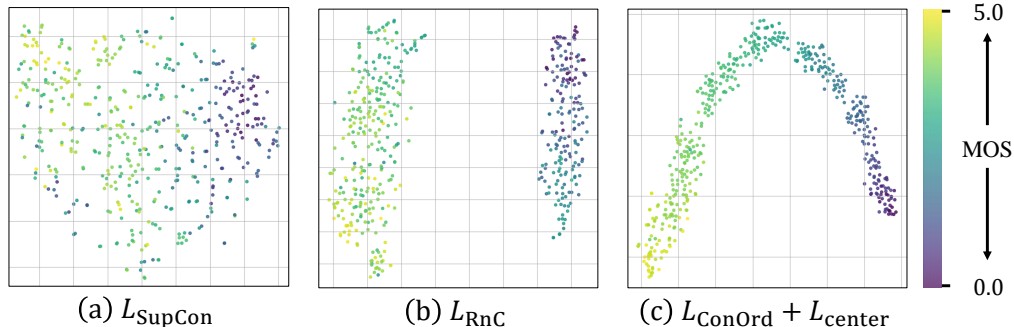

(a) $L_{\text{SupCon}}$     (b) $L_{\text{RnC}}$     (c) $L_{\text{ConOrd}} + L_{\text{center}}$

*Figure 13.* t-SNE visualization of embeddings learned by SupCon, RnC, and ConOrd on the BID dataset. Colors denote MOS scores.

**Training stability and gradient dynamics:** Figure 14 compares the optimization behavior of the first eight weighting configurations listed in Table 6. All configurations show smooth, monotonic decreases in training loss without signs of instability or divergence, indicating that the ConOrd formulation remains robust across a wide range of affinity-disparity designs. The gradient norms differ moderately in scale but remain consistently bounded and settle quickly into steady ranges. These results confirm that all eight variants exhibit well-behaved gradients and stable training dynamics, demonstrating the resilience of the proposed contrastive order formulation.

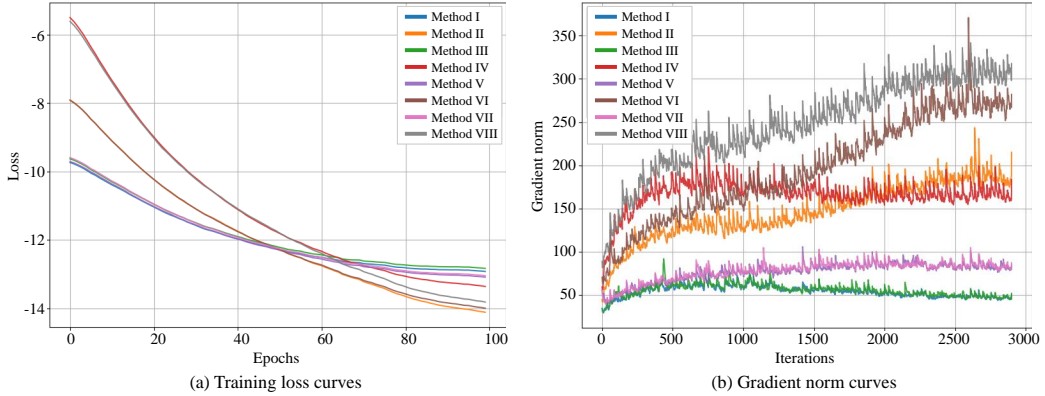

(a) Training loss curves     (b) Gradient norm curves

*Figure 14.* Training loss and gradient-norm dynamics across different weighting configurations, showing stable convergence and controlled gradients in all cases.

**Label frequency-aware weighting:** Incorporating label frequency can further improve performance in highly imbalanced settings. Letting $f_r$ denote the frequency of rank $r$, using frequency-aware reweighting reduces AgeDB-DIR MAE from 5.15 to 5.06, as shown in Table 23.

*Table 23.* Effect of label frequency-aware weighting on AgeDB-DIR.

| Method | $a_{ij}$ | $b_{ij}$ | AgeDB-DIR |
|---|---|---|---|
| Proposed | $\dfrac{1}{(r_i - r_j)^2 + \epsilon}$ | $(r_i - r_j)^2$ | 5.15 |
| Frequency-aware | $\dfrac{1}{(r_i - r_j)^2 + \epsilon} \cdot \dfrac{1}{\sqrt{f_{r_i} f_{r_j}}}$ | $(r_i - r_j)^2 \cdot \sqrt{f_{r_i} f_{r_j}}$ | **5.06** |

**Beyond $k$-NN inference:** Since ConOrd learns a metric-structured ordinal embedding space, $k$-NN serves as a natural nonparametric readout rather than a disconnected inference rule. However, ConOrd is not restricted to $k$-NN inference and also performs well with classification or regression heads, as shown in Table 24.

*Table 24.* Compatibility of ConOrd with classification and regression heads.

| Method | CLAP2015 MAE ($\downarrow$) | BID SRCC ($\uparrow$) | BID PCC ($\uparrow$) | LSVQ-test SRCC ($\uparrow$) | LSVQ-test PCC ($\uparrow$) |
|---|---|---|---|---|---|
| $L_{\text{CE}}$ | 2.605 | 0.742 | 0.762 | 0.902 | 0.901 |
| $L_{\text{CE}} + L_{\text{ConOrd}} + L_{\text{center}}$ | 2.599 | 0.876 | 0.881 | **0.904** | 0.903 |
| $L_{\text{MAE}}$ | 2.506 | 0.881 | 0.900 | 0.903 | 0.902 |
| $L_{\text{MAE}} + L_{\text{ConOrd}} + L_{\text{center}}$ | **2.481** | **0.899** | **0.914** | **0.904** | **0.904** |

**Effectiveness as an auxiliary loss:** We further examine whether ConOrd can be used as a plug-and-play auxiliary loss for an existing ordinal regression model. Specifically, we add $L_{\text{ConOrd}}$ to LoDa(Xu et al., 2024) without architectural modifications. As shown in Table 25, adding ConOrd improves both SRCC and PCC on the CLIVE dataset, demonstrating that the proposed loss can complement existing task-specific objectives.

*Table 25.* Effectiveness of ConOrd as an auxiliary loss on LoDa(Xu et al., 2024) for the CLIVE dataset.

| Method | SRCC ($\uparrow$) | PCC ($\uparrow$) |
|---|---|---|
| $L_{\text{plcc}}$ (Original LoDa) | 0.876 | 0.896 |
| $L_{\text{plcc}} + L_{\text{ConOrd}}$ | **0.880** | **0.900** |

**Subgroup analysis:** We conduct subgroup analysis on the AgeDB-DIR test set to examine potential demographic bias. As shown in Table 26, ConOrd shows comparable performance across gender groups, with slightly higher MAE for female samples. Across age groups, the MAE is higher for the youngest and oldest groups, particularly for the 80+ group, likely due to fewer samples and larger appearance variation.

*Table 26.* Subgroup analysis on the AgeDB-DIR test set.

| Analysis | Group | # samples | MAE ($\downarrow$) |
|---|---|---|---|
| Gender | Female | 867 | 5.51 |
| Gender | Male | 1273 | 4.90 |
| Age | 0–19 | 158 | 5.70 |
| Age | 20–39 | 600 | 4.50 |
| Age | 40–59 | 600 | 5.09 |
| Age | 60–79 | 595 | 5.19 |
| Age | 80+ | 187 | 6.77 |

## C.5. Complexity

We use PyTorch and NVIDIA GeForce RTX 4090 GPUs for all experiments.

**Training time:** Table 27 compares the average training times required for training one epoch on the SPAQ dataset. The proposed algorithm achieves the fastest training time per epoch. This efficiency is attributed to its design, which eliminates the need for data augmentation and pairwise sample construction. Unlike $L_{\text{SupCon}}$ and $L_{\text{RnC}}$, which should generate augmented sample pairs during training, the proposed ConOrd eliminates this step and improves efficiency. The RnC loss incurs the longest training time because it needs to dynamically select negative samples based on label distances for each anchor-positive pair. This conditional filtering introduces computational overhead and hinders parallelization. In contrast, ConOrd uses fixed weight masks, allowing more efficient and parallel computations. Thus, the proposed ConOrd loss requires the shortest computation time.

*Table 27.* Comparison of processing times required for training one epoch on SPAQ.

| Algorithm | Time (s) |
|---|---|
| $L_{\text{SupCon}}$ in (1) | 41.7 |
| $L_{\text{RnC}}$ in (7) | 65.5 |
| $L_{\text{ConOrd}} + L_{\text{center}}$ in (9) | 31.8 |

**Testing time on SPAQ:** We also report the average testing time on the SPAQ dataset. The whole process takes only $5.0 \times 10^{-3}$ seconds to test an image on average: 10.0 seconds for the full test feature extraction, and $1.5 \times 10^{-4}$ seconds for the score estimation. Hence, ConOrd provides a computationally efficient solution for practical deployment.

**Testing time on CLAP2015, BID, and LSVQ-test:** To further evaluate the reliability of the $k$-NN inference, we measure the end-to-end test-time latency for processing each full test dataset. We repeat this measurement over multiple runs and report the mean and standard deviation in Table 28. Compared with the state-of-the-art baselines for each task — NumCLIP for age estimation, LoDa for BIQA, and DOVER for BVQA — ConOrd achieves consistently faster and stable test-time performance across all datasets. While the $k$-NN inference complexity grows linearly with the training set size, the empirical results in Table 28 indicate that test-time latency remains low and stable at practical dataset scales.

*Table 28.* Mean and standard deviation of end-to-end test-time latency, measuring the variability of $k$-NN inference, on the CLAP2015, BID, and LSVQ-test datasets.

| Method | CLAP2015 (Age) | BID (BIQA) | LSVQ-test (BVQA) |
|---|---|---|---|
| SOTA baseline | NumCLIP: $5.15 \pm 0.42$s | LoDa: $5.63 \pm 0.31$s | DOVER: $1222.36 \pm 9.04$s |
| ConOrd ($k$-NN) | $2.16 \pm 0.14$s | $0.88 \pm 0.05$s | $253.70 \pm 7.65$s |

**Loss complexity:** To provide a clearer comparison of efficiency, we report the per-batch loss computation time and the GPU memory usage associated with computing $L_{\text{ConOrd}}$, $L_{\text{RnC}}$, and the BIQA loss used in QCN. As summarized in Tables 29 and 30, $L_{\text{ConOrd}}$ is the most computationally efficient in terms of runtime, while also maintaining low memory usage compared to prior methods.

*Table 29.* Loss-level computation time on BID (batch size = 32).

| Method | $L_{\text{ConOrd}}$ | $L_{\text{RnC}}$ | QCN |
|---|---|---|---|
| Loss computation time (batch size = 32) | 4.6ms | 8.7ms | 8.5ms |

*Table 30.* GPU memory usage for loss computation on BID.

| Algorithm | Memory |
|---|---|
| $L_{\text{ConOrd}}$ | 2.17MB |
| $L_{\text{RnC}}$ | 1.17MB |
| QCN | 273.78MB |

**Training time per epoch:** To further evaluate scalability beyond per-batch loss computation time, we report the training time per epoch on both the smaller BID dataset and the larger KonIQ-10k dataset. As shown in Table 31, despite all-pair interactions, ConOrd has training cost comparable to RnC (Zha et al., 2023) and is far more efficient than QCN (Shin et al., 2024), while scaling well to larger datasets.

*Table 31.* Training time per epoch on BID and KonIQ-10k.

| Dataset | $L_{ConOrd}$ | $L_{RnC}$ | QCN (Shin et al., 2024) |
|---|---|---|---|
| BID | 5.69s | 5.65s | 46.7s |
| KonIQ-10k | 24.5s | 27.5s | 287.4s |

**Model efficiency:** Table 32 compares the model complexity of the proposed BIQA algorithm with those of other recent algorithms. The proposed algorithm adopts ViT-B as the encoder, resulting in a complexity of 86M. While this is not the smallest among the compared models, the proposed algorithm consistently outperforms others across multiple BIQA benchmarks. In particular, it achieves better performance than LoDa, which uses a larger model, and LQMamba, which employs the same encoder architecture.

*Table 32.* Comparison with BIQA algorithms in terms of network complexity.

| Algorithm | # parameters (M) |
|---|---|
| ReIQA (Saha et al., 2023) | 47 |
| LQMamba (Guan et al., 2024) | 86 |
| QCN (Shin et al., 2024) | 30 |
| LoDa (Xu et al., 2024) | 95 |
| ConOrd (Proposed) | 86 |

# D. Regression Results

Figures 15, 16, and 17 show regression results of the proposed ConOrd on the facial age estimation, BIQA, and BVQA tasks, respectively.

## D.1. Facial Age Estimation

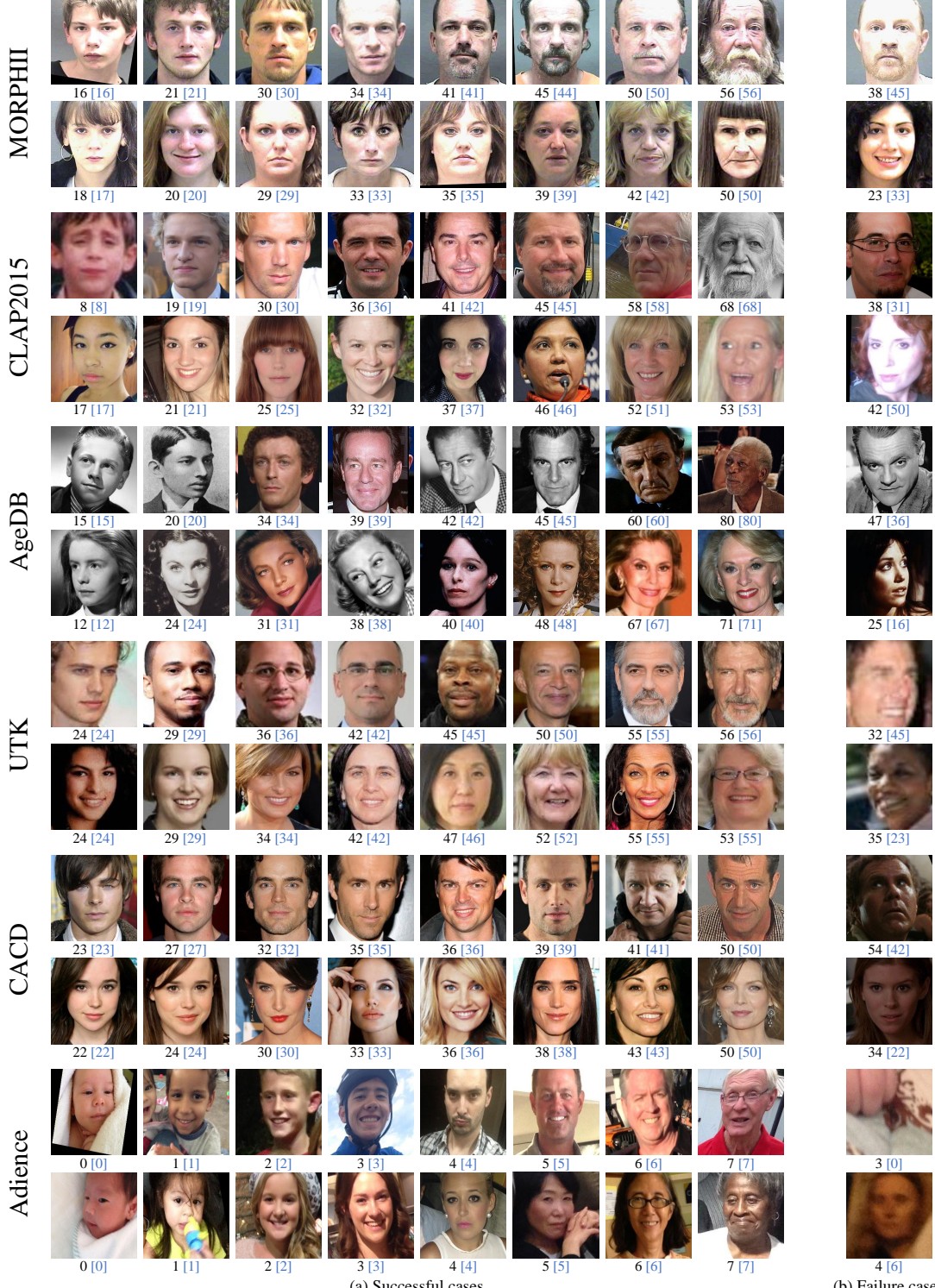

*Figure 15.* (a) Success and (b) failure cases of regression results on the facial age estimation datasets. Under each image, the estimated age is specified with the ground-truth in brackets.

## D.2. Blind Image Quality Assessment

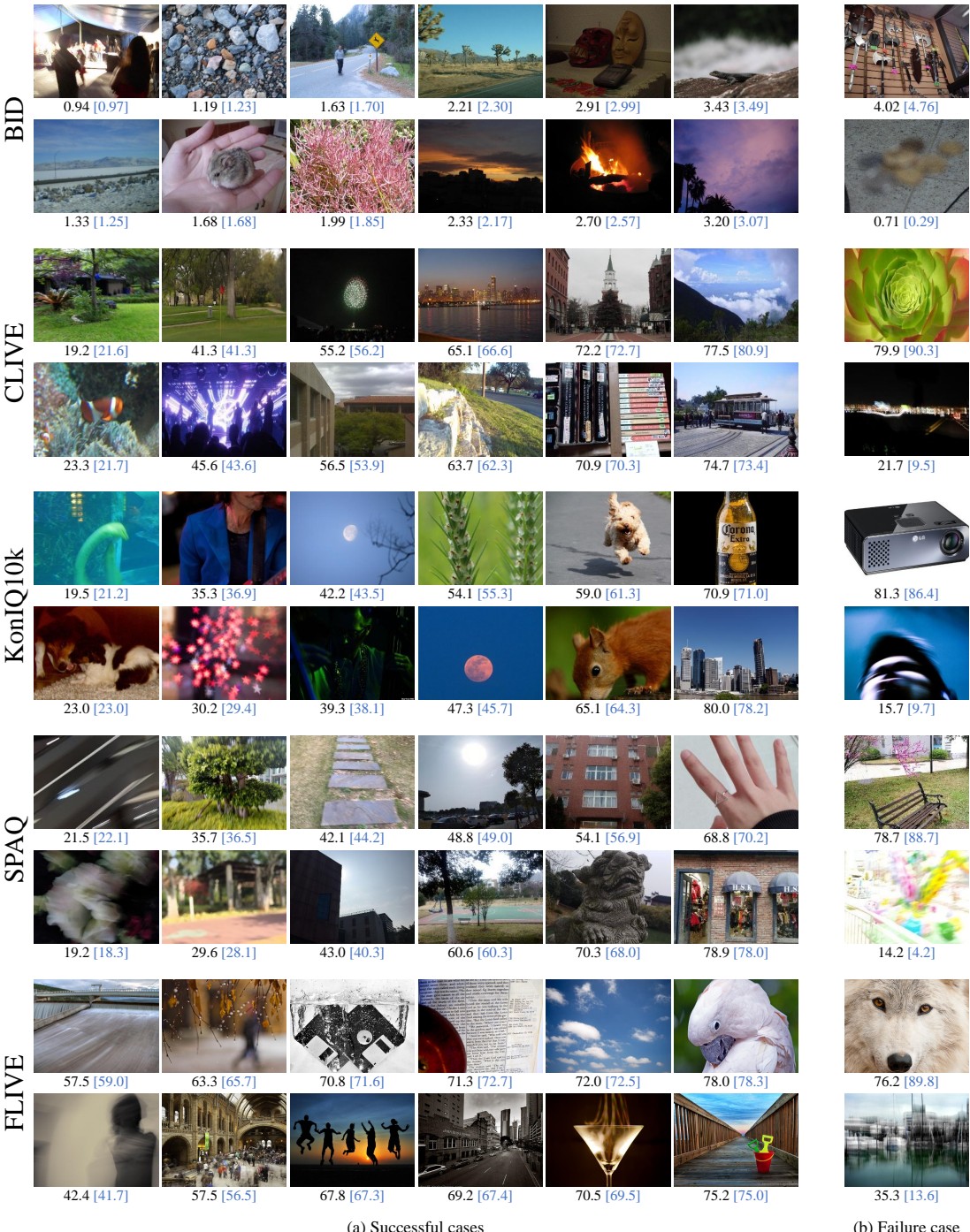

*Figure 16.* (a) Success and (b) failure cases of regression results on the BIQA datasets. Under each image, the estimated quality score is specified with the ground-truth in brackets.

## D.3. Blind Video Quality Assessment

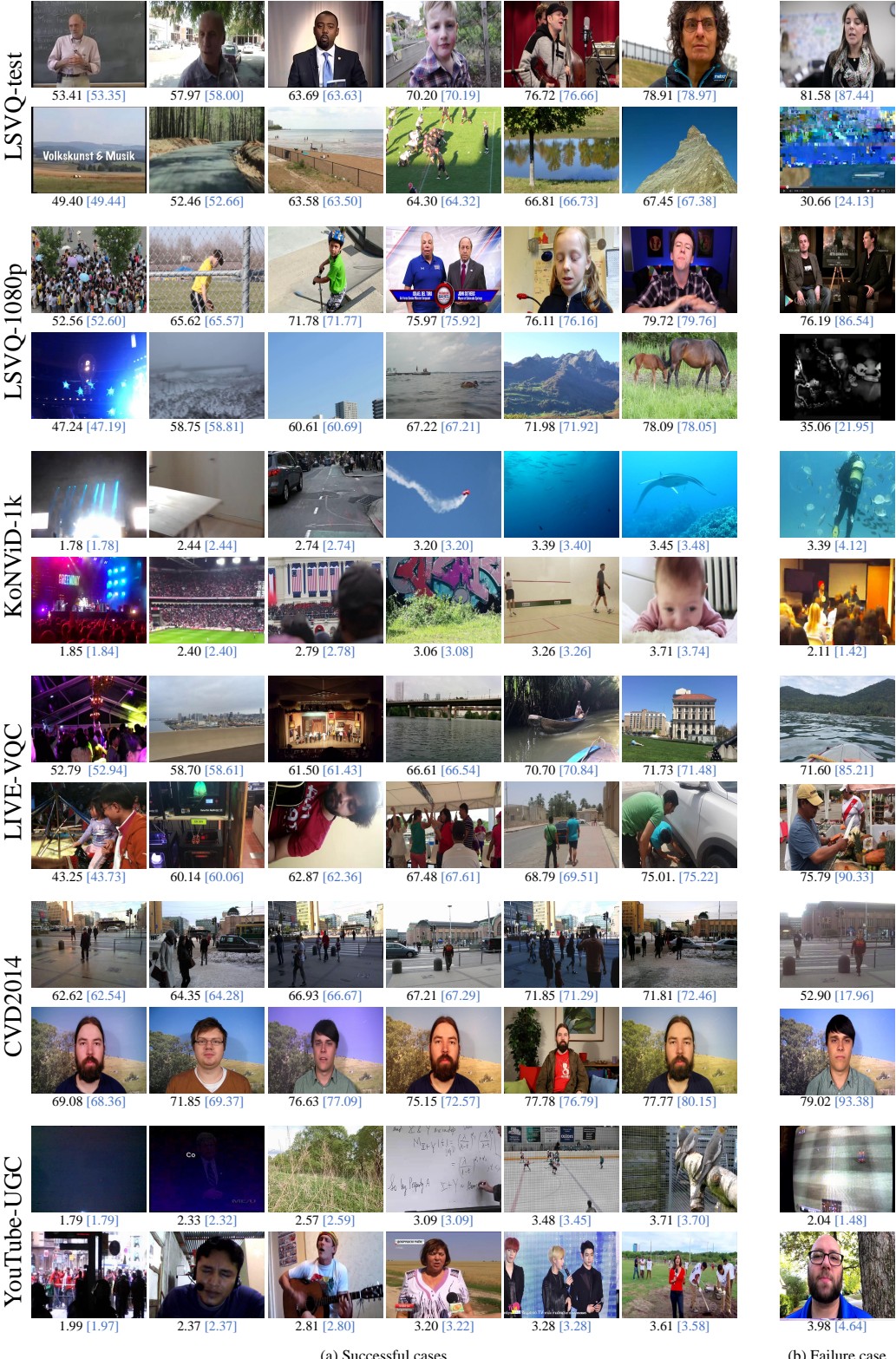

Figure 17. (a) Success and (b) failure cases of regression results on the BVQA datasets. Under each image, the estimated quality score is specified with the ground-truth in brackets.

# E. Limitations

As shown in Appendix D, the proposed algorithm generally demonstrates strong predictive performance across a variety of regression tasks and dataset types. However, along with successful cases, failure cases are also illustrated in Figures 15(b), 16(b), and 17(b). It is observed from Figure 15(b) that for the task of facial age estimation, the model may under- or over-estimate ages when age-related visual cues are ambiguous or degraded, such as under atypical lighting conditions, strong shadows, or reduced texture contrast. From the BIQA and BVQA results in Figures 16(b) and 17(b), we observe that prediction errors are more frequent when MOS values are extremely low or high. This appears to stem from the limited representation of such samples in the training data, suggesting that the performance could be further improved by adopting learning strategies that better handle imbalanced data distributions.

