# OpenReview forum: "Contrastive Order Learning: A General Framework for Ordinal Regression"
_ICML.cc/2026/Conference — ICML 2026 regular_

### Official Review · Reviewer_AXys · 2026-03-04

**Soundness:** 2
**Presentation:** 3
**Significance:** 2
**Originality:** 2
**Overall Recommendation:** 3
**Confidence:** 5

**Summary:**

This paper studies ordinal regression and proposes a soft-margin contrastive learning framework that leverages distance-aware negative sampling to encode ordinal relationships. The method is designed to improve representation learning for ordinal targets via a contrastive objective with an ordinal-aware margin.

**Compliance With Llm Reviewing Policy:**

Affirmed.

**Final Justification:**

I agree with Reviewer ogaZ’s assessment. As a researcher familiar with DIR-related methods, I believe the paper still has some fundamental issues that have not been sufficiently addressed.

My remaining concern mainly lies in the absence of evaluations on standard benchmarks such as IMDB-WIKI, STS, and NYUD2 depth, which are widely adopted in the DIR / deep regression literature. Without results on these datasets, it remains difficult to fully assess the general effectiveness of the proposed method.

**Key Questions For Authors:**

1.How does the proposed soft-margin distance-aware contrastive objective differ mathematically and empirically from prior ordinal-/distance-aware contrastive methods (e.g., Rank-n-Contrast / RankSim / ConR / VIR)? Can the authors provide a clearer isolated contribution?

2.Can the authors report results with matched backbones (e.g., ResNet-50/18 and/or ViT-B) across key baselines under the same training protocol to ensure a fair comparison (like Age estimation in Table1)?

3.Can the authors additionally evaluate on more established ordinal/DIR benchmarks (e.g., AgeDB, IMDB-WIKI, STS, NYUD2) to support the “general framework” claim?

4.For IQA/VQA experiments, which baselines are true ordinal regression baselines under the same architecture/training setup? If task-specific modules are used, how should we interpret the contribution of the proposed ordinal contrastive loss?

Question 2 to 4 are all related to fair experiment comparisions and need careful considerations.

**Limitations:**

The authors have discussed the technical limitations of the proposed methods in supplementary materials.

**Strengths And Weaknesses:**

Strengths:

The formulation is conceptually straightforward, and the paper provides clear definitions and implementation details for the proposed soft-margin contrastive learning objective, making the approach easy to follow and reproduce.

Weaknesses:

1.Limited novelty in motivation and technical contribution.

The paper emphasizes the importance of distance-aware (ordinal-aware) negative samples and margins in contrastive learning for ordinal regression. However, similar motivations and core ideas—i.e., explicitly modeling label order/distance in representation learning—have been discussed and explored in a number of prior works (e.g., AdaCon, Ordinal Entropy, Rank-n-Contrast, RankSim, ConR, VIR, SRL, among others). As a result, the novelty of the main idea and the motivation as a primary contribution seems limited. It would help if the authors could more explicitly discuss what is fundamentally new beyond existing ordinal-/distance-aware contrastive formulations.

2.Potentially unfair baseline comparisons due to backbone mismatch (most important issue).

In Table 1 (age estimation), many standard DIR/ordinal regression baselines are typically evaluated with ResNet-based backbones to isolate the contribution of the learning objective.
In contrast, this paper adopts ViT-B (motivated by CLIP-based pipelines) and compares against methods using different backbones. This makes it difficult to attribute performance gains to the proposed contrastive ordinal objective rather than to the backbone choice.

While the additional results in Table 8 with a ResNet18 backbone are useful, the improvement over RnC appears notably smaller than the gains reported in Table 1, which raises concerns that the main-paper comparison may overstate the benefit of the proposed method.
For a fair assessment, the authors should provide comparisons under matched backbones and training protocols across representative ordinal regression baselines.

3.Evaluation tasks and baselines are not well-aligned with the “general ordinal regression” claim.

The inclusion of IQA/VQA datasets may be less convincing for positioning the method as a general ordinal regression framework, since these tasks often require task-specific architectures or modules, and it is unclear whether standard ordinal regression baselines are re-implemented under comparable settings in Tables 2–3.
Given the paper’s position as a fundamental ordinal regression method, it would be more compelling to evaluate on widely used ordinal/DIR benchmarks (e.g., AgeDB, IMDB-WIKI, STS, NYUD2 depth) and include strong representative ordinal regression baselines under a consistent protocol. Otherwise, it remains hard to isolate the effect of the proposed contrastive module versus generic gains from adding naive contrastive learning, supercon, etc.

---

> ### Author Rebuttal · Authors · 2026-03-30
>
> We appreciate the reviewer’s thorough evaluation and constructive suggestions. We address each concern below and will incorporate the corresponding clarifications and additional results into the revised manuscript.
>
> ***
> > **(Q1) Clarifying the isolated contribution relative to prior ordinal-aware methods**
>
> Relative to prior ordinal-aware methods, including RnC, RankSim, ConR, and VIR, the key distinction of ConOrd can be summarized along three aspects: (A) loss formulation, (B) supervision and gradient behavior, and (C) pairwise coverage.
>
> **(A) Loss formulation**
> - RankSim and VIR are not contrastive methods.
> - RnC and ConR are contrastive, but use threshold-based binary pair construction.
> - ConOrd instead uses continuous rank-dependent weighting, directly incorporating ordinal distance into the loss.
>
> **(B) Supervision and gradient behavior**
> - RankSim relies on ranking constraints whose gradients diminish once satisfied.
> - RnC and ConR capture ordering, but not the magnitude of rank differences.
> - VIR does not impose pairwise ordinal supervision in a contrastive form.
> - ConOrd uses explicit rank-difference-dependent attraction and repulsion, yielding distance-aware gradients.
>
> **(C) Pairwise coverage**
> - RnC and ConR rely on thresholded positive/negative construction.
> - RankSim does not provide uniformly dense supervision because satisfied constraints contribute little or no gradient.
> - ConOrd assigns non-binary weights to all pairs in the batch, providing denser ordinal supervision.
>
> Overall, prior methods mainly rely on discrete or partially active ordinal supervision, whereas ConOrd provides continuous, distance-aware pairwise weighting over all pairs within the batch.
>
> &nbsp;
>
> > **(Q3) Additional evaluation on established ordinal/DIR benchmarks**
>
> To further support this distinction empirically, we additionally evaluate ConOrd on the reviewer-mentioned ordinal regression benchmarks, including AgeDB and IMDB-WIKI. Under the same ResNet-50 backbone, ConOrd outperforms prior ordinal-aware methods. Experiments on STS-B and NYUD2 are also currently in progress, and we will provide those results as they become available.
>
> |Method|AgeDB|IMDB-WIKI|
> |-|-|-|
> ||MAE($\downarrow$)|MAE($\downarrow$)|
> |RankSim|7.02|7.50|
> |RnC|6.10|-|
> |VIR|6.99|7.19|
> |ConR|7.20|7.33|
> |ConOrd (Proposed)|**6.02**|**7.06**|
>
> &nbsp;
>
> > **(Q2+Q4) Controlled comparisons under matched backbones for IQA/VQA**
>
> In Table 4, all compared methods already use the same backbone (ViT-B), the same training protocol, and the same k-NN inference scheme. Therefore, the performance differences are attributable to the loss formulation rather than architectural or training discrepancies. We will clarify this controlled setup more explicitly in the revised manuscript.
>
> We further extend the same controlled setting to BIQA (BID), in addition to the original comparisons on age estimation (CLAP2015) and BVQA (LSVQ-1080p). Under this matched setup, ConOrd achieves the best performance across age estimation, BIQA, and BVQA.
>
> Therefore, these IQA/VQA results should be interpreted as controlled evidence for the contribution of the proposed ordinal contrastive loss under the same model and evaluation pipeline, rather than as architecture-level comparisons.
>
> |Method|CLAP2015|BID|BID|LSVQ-1080p|LSVQ-1080p|
> |-|-|-|-|-|-|
> ||MAE($\downarrow$)|SRCC($\uparrow$)|PCC($\uparrow$)|SRCC($\uparrow$)|PCC ($\uparrow$)|
> |$L_{\text{SupCon}}$|2.625|0.819|0.876|0.614|0.682|
> |$L_{\text{OCL}}$|2.597|0.909|0.921|0.697|0.687|
> |$L_{\text{MMNP}}+L_{\text{CE}}$|2.777|0.828|0.825|0.716|0.727|
> |$L_{\text{RnC}}$|2.531|0.892|0.906|0.812|0.843|
> |$L_{\text{ConOrd}}+L_{\text{center}}$ (Proposed)|**2.461**|**0.913**|**0.925**|**0.818**|**0.851**|
>
> &nbsp;
>
> > **Why the inclusion of IQA/VQA is appropriate**
>
> We respectfully disagree with the concern that the inclusion of IQA/VQA is less convincing for evaluating a general ordinal regression framework. IQA and VQA are standard ordinal regression tasks with clear label ordering, and therefore constitute meaningful testbeds for assessing whether an ordinal loss generalizes beyond a single benchmark family.
>
> Our goal in including IQA/VQA is not to claim task-specific architectural novelty in those domains, but to examine whether the proposed ordinal contrastive loss remains beneficial in realistic ordinal regression settings, including those that commonly use stronger backbones or task-specific pipelines. Rather than weakening the general framework claim, these experiments broaden the evidence for it.
>
>
> ***
> If you have additional comments, please let us know. We will do our best to address them. Thank you again for your thoughtful feedback.

---

> > ### Author Rebuttal · Reviewer_AXys · 2026-04-02
> >
> > Thank you for the rebuttal. I still have several questions.
> >
> > Regarding the comparison with prior work (e.g., Ordinal Entropy, Rank-n-Contrast, RankSim, ConR, VIR, SRL), I noticed that only Table 1 partially includes some strong deep regression baselines. However, such comparisons are missing in Tables 2 and 3.
> >
> > I would encourage the authors to more comprehensively review the DIR / deep regression literature, as many of these methods are evaluated on standard benchmarks (e.g., IMDB-WIKI, STS, NYUD2 depth) under comparable settings, enabling fair comparisons. While I do not mean to suggest that the IQA dataset used in this work is not meaningful, the absence of evaluations on these commonly used datasets, together with missing comparisons to established baselines, makes it difficult to adequately assess the effectiveness of the proposed method. This remains my primary concern.
> >
> > Additionally, regarding the results provided in Q3, I would like to clarify whether the reported numbers are based on the AgeDB-DIR setting or the original AgeDB setting. The MAE can differ significantly between these two protocols, and mixing them may lead to unfair comparisons.

---

> > > ### Author Response · Authors · 2026-04-03
> > >
> > > We sincerely thank you for your additional feedback and for carefully reviewing our rebuttal.
> > >
> > > ***
> > >
> > > > **Comparison to deep regression methods for IQA/VQA (for Tables 2 and 3)**
> > >
> > > To directly address your concern, we additionally compare ConOrd with representative DIR / deep regression methods (e.g., RankSim, Ordinal Entropy, Rank-n-Contrast, ConR, SRL) on IQA/VQA benchmarks. To ensure a fair comparison, all methods are implemented under the same backbone, training protocol, and evaluation pipeline as ConOrd. As shown in the table below, ConOrd consistently outperforms all compared deep regression methods. We will include these additional comparisons and clearly describe the unified evaluation protocol in the revised manuscript.
> > >
> > > |Method|BID|BID|LSVQ-1080p|LSVQ-1080p|
> > > |-|-|-|-|-|
> > > ||SRCC($\uparrow$)|PCC($\uparrow$)|SRCC($\uparrow$)|PCC($\uparrow$)|
> > > |RankSim|0.892|0.909|0.814|0.848|
> > > |Ordinal Entropy|0.886|0.909|0.809|0.845|
> > > |Rank-n-Contrast|0.892|0.906|0.812|0.843|
> > > |ConR|0.885|0.909|0.812|0.846|
> > > |SRL|0.900|0.921|0.814|0.847|
> > > |ConOrd (Proposed)|**0.913**|**0.925**|**0.818**|**0.851**|
> > >
> > >
> > > &nbsp;
> > >
> > > > **Clarification on the AgeDB dataset for the results in Q3**
> > >
> > > We appreciate your careful observation. The results reported in Q3 are based on the **AgeDB-DIR** setting, and our dataset configuration matches the standard AgeDB-DIR protocol. We agree that the notation may cause ambiguity, and we will explicitly denote this protocol as **AgeDB-DIR** throughout the revised paper.
> > >
> > > &nbsp;
> > >
> > > > **Updated table for Q3**
> > >
> > > We further include STS-B-DIR and NYUD2-DIR results on table for Q3. Following prior work, we use a BiLSTM with GloVe embeddings for STS-B-DIR and a ResNet-50-based encoder–decoder [1] for NYUD2-DIR.
> > >
> > > |Method|AgeDB-DIR|IMDB-WIKI-DIR|STS-B-DIR|NYUD2-DIR|
> > > |-|-|-|-|-|
> > > ||MAE($\downarrow$)|MAE($\downarrow$)|MSE($\downarrow$)|RMSE($\downarrow$)
> > > |RankSim|7.02|7.50|0.903|-|
> > > |Ordinal Entropy|7.46|-|-|-|
> > > |Rank-n-Contrast|6.14|-|-|-|
> > > |ConR|7.20|7.33|-|1.304|
> > > |VIR|6.99|7.19|0.892|1.305|
> > > |SRL|7.22|7.69|0.877|-|
> > > |ConOrd (Proposed)|**6.02**|**7.06**|**0.868**|**1.294**|
> > >
> > > [1] Hu et al. "Revisiting single image depth estimation: Toward higher resolution maps with accurate object boundaries," WACV 2019.
> > >
> > > ***
> > > Thank you again for your constructive feedback. It has helped us improve the clarity and completeness of our work.

---

### Official Review · Reviewer_Hphf · 2026-03-09

**Soundness:** 3
**Presentation:** 3
**Significance:** 3
**Originality:** 2
**Overall Recommendation:** 4
**Confidence:** 5

**Summary:**

To better handle order learning for certain tasks, e.g., image quality assessment, efforts are made upon the loss functions. This paper proposes a framework, combing contrastive learning and a designed order loss function, to encourage better attraction between samples with ranks, for minor or large gaps. Supported by corresponding theories and validations. Experiments verify the effectiveness and demonstrate competitive performance.

**Compliance With Llm Reviewing Policy:**

Affirmed.

**Final Justification:**

Good to see the rebuttal. The rating is kept.

**Key Questions For Authors:**

See Major Weaknesses. The reviewer's current rating of \<Weak accept\> is contingent on a satisfactory rebuttal.

**Limitations:**

This paper contains sections discussing limitations of the work, along with a section to state the societal impact.

**Strengths And Weaknesses:**

[Strengths]
1. The motivation of this paper is clear and well justified. The core idea is simple yet effective.
2. Theoretical analysis is provided within this paper for reference.
3. The experimental validation is comprehensive and extensive.
4. Presentations are good, and the paper is easy to follow.

[Major Weaknesses]
1. The reviewer finds that, while the paper specifies particular forms for $a_{ij}$ and $b_{ij}$ and justifies them as “simple, smooth, symmetric, and strictly monotonic functions.” However, the latter statement describes as that ConOrd is not sensitive to the specific functions. Does this imply that the choice of $a_{ij}$ and $b_{ij}$ is empirical?
2. In Supp Sec A, the authors provide a gradient analysis showing that the factor has regional patterns. Given this, however, discussions are needed as whether the method can correct if with severe misalignment in the early stage of training. For example, if the model initially maps some truly relevant samples far apart, those pairs may later receive insufficient gradients to be pulled back together.
3. The ConOrd loss performs pairwise comparisons within a batch, thus $O(B^2)$ complexity. Since the paper states that the idea contributes to a general framework, is it still practical, i.e., if the dataset is large? Consider providing training loss or profiling details for reference.

[Minor Weaknesses]
1. The authors describe ConOrd as a relaxation of RnC, and the final loss is simply a combination. Such renders the paper incremental, more like an adaptation of existing approaches than a fundamentally new learning paradigm. In that sense, the title may feel a bit overstated.
2. The references are rather limited, especially for IQA, consider adding more works to it.

---

> ### Author Rebuttal · Authors · 2026-03-29
>
> Thank you for your positive and constructive evaluation of our work. Please find our responses below, which will be incorporated into the revised paper.
>
> ***
>
> > **Weighting design**
>
> We agree that the quadratic form is not the only possible choice. However, it provides a simple instantiation that satisfies a key monotonic transition condition: as the rank gap increases, the interaction shifts consistently from attraction to repulsion. The comparable performance of alternatives in Table 6 therefore suggests that the benefit comes from this continuous distance-aware mechanism itself, rather than from a specific formula. We have additionally conducted a more detailed gradient-based analysis of this point and will incorporate it into the revised manuscript.
>
> &nbsp;
>
> > **Early misalignment**
>
> ConOrd is explicitly designed to mitigate early-stage misalignment through the following mechanisms.
>
> - **Global correction via $L_{\text{center}}$:**
> Even if similar-rank samples are initially far apart and receive weak pairwise gradients, both are independently pulled toward the same reference point $\mu_r$, providing a consistent global signal for coarse alignment.
>
> - **All-pair interactions in $L_{\text{ConOrd}}$:**
> Unlike margin-based methods that rely on selected pairs, ConOrd considers all pairwise interactions, ensuring that corrective signals remain available even when some pairs are initially misaligned. In practice, training remains stable across settings.
>
> &nbsp;
>
> > **Complexity**
>
> The per-batch computation time is reported in Appendix C.4. To further evaluate scalability, we additionally report training time per epoch on both the smaller BID dataset and the larger KonIQ-10k dataset. These results show that, despite all-pair interactions, ConOrd has training cost comparable to RnC and is far more efficient than QCN, while scaling well to larger datasets.
>
> |Dataset|Metric|$L_{\text{ConOrd}}$|$L_{\text{RnC}}$|QCN|
> |-|-|-|-|-|
> |BID|Training time (1 epoch) | 5.69s | 5.65s | 46.7s |
> |KonIQ-10k|Training time (1 epoch)| 24.5s | 27.5s | 287.4s |
>
> &nbsp;
>
> > **Title and positioning of the framework**
>
> We agree that ConOrd is built from familiar ingredients such as contrastive objectives, embedding-based learning, regularization, and k-NN inference. Our intention in the title was therefore not to claim an entirely new learning paradigm, but to emphasize that ConOrd provides a general framework for integrating the complementary strengths of contrastive learning and order learning in ordinal regression. In particular, the proposed objective replaces threshold-based or partially active supervision with continuous, distance-aware, all-pair supervision, allowing the model to exploit the full ordinal spectrum within a batch. We will clarify this positioning more explicitly in the revised manuscript so that the title is interpreted in this intended sense.
>
> &nbsp;
>
> > **IQA references**
>
> Following your suggestion, we include additional recent IQA methods for a more comprehensive comparison. As shown below, ConOrd achieves the best performance across all benchmarks. This extended comparison will be included in the revised manuscript.
>
> |Algorithm| BID (SRCC) | BID (PCC) | CLIVE (SRCC) | CLIVE (PCC) | KonIQ10k (SRCC) | KonIQ10k (PCC) | SPAQ (SRCC) | SPAQ (PCC) | FLIVE (SRCC) | FLIVE (PCC) |
> |-|-|-|-|-|-|-|-|-|-|-|
> |TOPIQ (TIP 2024)|-|-| 0.870 | 0.884 | 0.926 | 0.933 | - | - | 0.633 | 0.722 |
> |SaTQA (AAAI 2024)|-|-| 0.877 | 0.903 | 0.930 | 0.941 | - | - | 0.582 | 0.676 |
> |SS-IQA (AAAI 2024)|-|-| 0.835 | 0.869 | 0.913 | 0.932 | 0.826 | 0.824 | 0.542 | 0.582 |
> |RichIQA (TCSVT 2025)| 0.900 | 0.909 | 0.894 | 0.912 | 0.938 | 0.950 | 0.923 | 0.929 | 0.583 | 0.684 |
> |UNQA (TCSVT 2025)| 0.881 | 0.914 | 0.874 | 0.903 | 0.893 | 0.915 | 0.913 | 0.917 | - | - |
> |ConOrd (Proposed)| **0.913** | **0.925** | **0.900** | **0.921** | **0.947** | **0.958** | **0.927** | **0.931** | **0.651** | **0.752** |
>
> ***
> We sincerely thank the reviewer again for the positive and constructive feedback.

---

> > ### Author Rebuttal · Reviewer_Hphf · 2026-04-03
> >
> > Thanks for the rebuttal. Please include these more details in the revision. The rating is kept.

---

### Official Review · Reviewer_FSjq · 2026-03-12

**Soundness:** 3
**Presentation:** 3
**Significance:** 2
**Originality:** 3
**Overall Recommendation:** 5
**Confidence:** 5

**Summary:**

The paper proposes Contrastive Order Learning (ConOrd), a contrastive framework that models ordinal relationships using soft affinity and disparity weights derived from rank differences. Instead of defining binary positive/negative pairs, ConOrd performs all-pair comparisons within a batch, where attraction and repulsion strengths depend on the ordinal distance between labels. The method is implemented as a contrastive order loss combined with a center loss to structure the embedding space. Experiments on multiple ordinal regression tasks,ncluding facial age estimation, blind image quality assessment, and blind video quality assessment, showing consistent improvements over prior methods such as GOL and Rank-N-Contrast.

**Compliance With Llm Reviewing Policy:**

Affirmed.

**Key Questions For Authors:**

See Weaknesses Above.

**Strengths And Weaknesses:**

Strengths:

1. The paper identifies a real limitation in existing methods: contrastive learning treats ordinal labels categorically, while order learning relies on local pairwise margins. The proposed formulation bridges these paradigms in a principled way.

2. The contrastive order loss introduces soft weighting based on rank gaps, enabling continuous ordinal supervision across all pairs. The formulation is straightforward and easy to implement.

3. Good ablation and analysis. The paper includes comparisons with alternative contrastive losses, ablations on affinity functions, and embedding visualizations, which help support the proposed design.

5. The framework is task-agnostic and can be applied to various ordinal regression settings, making it potentially useful beyond the specific tasks studied.

Weaknesses

1. The method is conceptually close to Rank-N-Contrast and other ordinal contrastive objectives. The main difference lies in replacing hard thresholds with soft weighting functions, which may be viewed as an incremental modification rather than a fundamentally new framework.

2. While the paper provides gradient analysis, it lacks deeper theoretical justification (e.g., guarantees about ordinal consistency or convergence properties). The design of the affinity/disparity functions appears largely heuristic.

3. Many baselines differ in architecture or pretraining strategy (e.g., transformer backbones), making it difficult to attribute gains solely to the proposed loss.

---

> ### Author Rebuttal · Authors · 2026-03-29
>
> We sincerely thank the reviewer for the encouraging and positive feedback. Below, we clarify the remaining points and will incorporate the corresponding revisions into the manuscript.
>
> ***
>
> > **Novelty**
>
> We agree that ConOrd is conceptually related to prior ordinal contrastive methods, but our novelty claim is not that each component is individually new. Rather, the proposed objective introduces a different formulation for incorporating ordinal structure into contrastive learning: continuous, distance-aware, all-pair supervision, instead of threshold-based or partially active ordinal supervision. This difference allows ConOrd to exploit the full ordinal spectrum within a batch, rather than a subset of selected pairs. We will revise the manuscript to make this formulation-level distinction more explicit.
>
> &nbsp;
>
> > **Justification for the affinity and disparity weights**
>
> To further clarify the choice of the affinity/disparity weights, we additionally analyzed the gradient behavior of $L_{\text{ConOrd}}$. From Eq. (18), pair $(i,j)$ contributes attraction if $\frac{a_{ij}}{\alpha_i} > \frac{b_{ij}}{\beta_i}$, i.e., $\frac{a(d_{ij})}{b(d_{ij})} > \frac{\alpha_i}{\beta_i}$, and repulsion otherwise. Since $\frac{\alpha_i}{\beta_i}$ is a shared threshold for anchor $i$, a key condition for the desired ordinal behavior is that $\frac{a(d)}{b(d)}$ decreases monotonically with $d$. This induces a single crossover rank gap, below which pairs are encouraged to attract and above which they are encouraged to repel, yielding a consistent ordinal interaction pattern for each anchor.
>
> From this perspective, the different configurations in Table 6 can be interpreted according to whether they satisfy this monotone transition property. Weightings that satisfy it (Methods I–VIII) all perform comparably well, whereas configurations that break it (Methods IX and X) show degraded performance.
>
> |Category|Methods|MAE|Monotone Transition|
> |-|-|-|-|
> |Valid function class|I–VIII|2.461–2.518|o|
> |Coarse approximation|IX|2.536|x (discontinuous)|
> |Unconstrained|X|2.880|x (violates monotonicity)|
>
> Thus, the residual variation within Methods I–VIII is better interpreted as a secondary effect of how each valid function emphasizes different rank gaps (and the choice of $\kappa_{ij}$), rather than evidence that the quadratic form is uniquely special. We will incorporate this additional analysis into the revised manuscript.
>
> &nbsp;
>
> > **Backbone fairness**
>
> We would like to clarify that the comparisons in Table 4 were already conducted under a fully controlled setting, where
> - all methods share the same backbone (ViT-B),
> - identical training protocol is used across all baselines,
> - a unified k-NN inference scheme is adopted for evaluation.
>
> This design ensures that the performance differences are attributable solely to the loss formulation, rather than architectural or training discrepancies. However, we acknowledge that this setup was not explicitly stated in the current manuscript, which may have led to confusion. To address this, we will add this clarification in the revised manuscript.
>
> |Method|CLAP2015|LSVQ-1080p|LSVQ-1080p|
> |-|-|-|-|
> | |MAE|SRCC|PCC|
> |$L_{\text{SupCon}}$|2.625|0.614|0.682|
> |$L_{\text{OCL}}$|2.597|0.697|0.687|
> |$L_{\text{MMNP}}+L_{\text{CE}}$|2.777|0.716|0.727|
> |$L_{\text{RnC}}$|2.531|0.812|0.843|
> |$L_{\text{ConOrd}}+L_{\text{center}}$ (Proposed)|**2.461**|**0.818**|**0.851**|
>
> ***
> We sincerely thank the reviewer again for the encouraging and positive feedback.

---

> > ### Author Rebuttal · Reviewer_FSjq · 2026-04-06
> >
> > The authors addressed my concerns, hence I maintain my postive socore.

---

### Official Review · Reviewer_ogaZ · 2026-03-12

**Soundness:** 2
**Presentation:** 3
**Significance:** 3
**Originality:** 2
**Overall Recommendation:** 3
**Confidence:** 3

**Summary:**

This paper introduces Contrastive Order Learning (ConOrd), a framework for ordinal regression that combines the strengths of contrastive learning and order learning. The core idea is that standard contrastive learning treats labels categorically and ignores their natural ordering, while existing order learning methods rely on local, margin-based comparisons that miss global structure. ConOrd addresses both limitations by proposing a contrastive order loss that assigns soft affinity and disparity weights to all sample pairs in a batch based on their rank differences — samples with similar ranks are attracted and those with distant ranks are repelled, with the strength modulated continuously rather than through hard positive/negative assignments. The affinity weight decreases with the squared rank gap while the disparity weight increases, and an exponential kernel based on embedding distance focuses updates on the most informative pairs. The authors demonstrate state-of-the-art results across facial age estimation (six datasets), blind image quality assessment (five datasets), and blind video quality assessment (five datasets), consistently outperforming prior methods including RnC, GOL, and various CLIP-based approaches.

**Compliance With Llm Reviewing Policy:**

Affirmed.

**Final Justification:**

While the paper presents a contrastive learning framework for ordinal regression, I recommend a weak rejection due to limited theoretical novelty and significant flaws in the empirical evaluation. The mathematical and technical contributions offer limited novelty beyond existing distance-aware and contrastive ordinal regression methods. More critically, the evaluation lacks fairness because the proposed method is compared against standard regression approaches using mismatched backbones. To properly interpret the added value of the new objective function rather than the representational power of the backbone itself, comparisons must be conducted under strictly matched architectures and identical training protocols. Finally, the specific choice of evaluation datasets does not adequately support the authors' claim of proposing a "general" ordinal regression framework, leaving the comprehensive evaluation lacking and the fundamental efficacy of the method unproven.

**Key Questions For Authors:**

How does the framework perform under severe label imbalance, and can the soft weighting scheme be adapted to address this common challenge?

The authors acknowledge in the limitations section that prediction errors concentrate at extreme rank values due to underrepresentation in training data, yet no strategy is proposed to handle this. Since the affinity and disparity weights depend solely on rank gaps and do not account for label frequency, samples from underrepresented ranks may receive insufficient learning signal relative to those from densely populated regions. Have the authors considered incorporating label frequency information into the weighting scheme? For instance, upweighting contributions from rare ranks, and if so, how does this affect performance on datasets with known imbalance? This seems particularly important given that the authors cite SLACE as related work addressing this exact issue.


How general is the ConOrd framework beyond k-NN inference, and can it be effectively combined with other prediction heads or training objectives?

The paper claims to present a "general framework" for ordinal regression, but all experiments exclusively use k-NN inference with ℓ2-normalized embeddings. This ties the method to a specific inference paradigm that requires storing all training embeddings and tuning k per dataset. Could the authors demonstrate that the ConOrd loss also improves performance when paired with a linear regression head or an ordinal classification head trained end-to-end? Additionally, have you explored using ConOrd as an auxiliary loss alongside standard regression losses like MSE or L1, and if so, does the combination yield further gains? This would substantially strengthen the claim of generality.

**Limitations:**

Limitations (Appendix E): The authors identify two concrete failure modes from their qualitative results — age estimation errors under ambiguous visual cues (unusual lighting, shadows) and larger prediction errors at extreme MOS values in quality assessment due to underrepresentation in training data. While these observations are valid and grounded in the actual experimental results, they remain descriptive rather than analytical. The authors note that imbalanced data distributions may be a contributing factor but do not propose any concrete mitigation strategies, explore how the soft weighting scheme might be adapted to address this, or discuss whether the issue is fundamental to the ConOrd formulation or a data-level problem. There is also no discussion of other important limitations, such as the reliance on k-NN inference (which scales linearly with training set size and requires per-dataset tuning of k), the dependence on strong pretrained backbones, or the assumption that rank differences are semantically uniform across the label space, e.g., an assumption that may not hold in domains where the perceptual difference between, say, ages 5 and 10 is qualitatively different from that between ages 50 and 55.

Societal Impact: The impact statement is brief and generic. It correctly flags that facial attribute prediction can raise ethical concerns, particularly regarding bias in training data, and recommends using the model as a decision-support tool rather than an autonomous system. However, it does not engage with specific risks in any depth. For example, there is no discussion of demographic bias in age estimation (which is a well-documented issue), no analysis of whether ConOrd's performance varies across demographic groups, and no consideration of how the method might be misused in surveillance or discriminatory profiling contexts. The recommendation for "fair evaluations" is vague and does not reference any specific fairness metrics or evaluation protocols.

**Strengths And Weaknesses:**

Strengths:

The proposed ConOrd loss is well-grounded mathematically, with a thorough gradient analysis (Appendix A) that shows how the affinity and disparity weights induce attraction for similar ranks and repulsion for distant ones.

The paper is clearly written and well-structured, with effective use of figures to build intuition.

The experimental evaluation is remarkably comprehensive, spanning three distinct ordinal regression domains, i.e., facial age estimation, blind image quality assessment, and blind video quality assessment, across a total of more than 16 benchmarks.


Weaknesses:

The k-NN inference strategy (Equation 10) introduces a disconnect between training and testing: the model is trained with a contrastive loss operating on embeddings, but inference relies on a separate k-NN rule whose hyperparameter k varies significantly across datasets (from 4 to 60 in Table 7) without clear guidance on how to select it. While Table 11 suggests performance is relatively stable across k values, the wide range of optimal k across tasks raises questions about whether the embedding space is uniformly well-structured or whether k is compensating for varying degrees of clustering quality. A more principled connection between the training objective and the inference mechanism would strengthen the approach.

The choice of quadratic forms for the affinity and disparity weights (Equations 4 and 5) is described as the simplest smooth monotonic option, but the justification is largely empirical rather than theoretical. Table 6 shows that several alternative configurations perform comparably (e.g., methods VI and VII are within 0.01 MAE of method VIII), which suggests the method may not be highly sensitive to this choice but also raises the question of whether the specific functional form truly matters or whether any reasonable monotonic weighting would suffice. A more formal analysis of what properties the weighting functions must satisfy for convergence or optimality guarantees would be valuable.

The paper defers a significant amount of essential information to the appendices, including all dataset descriptions, implementation details, and training configurations. While space constraints are understandable, readers must consult the appendix to understand basic experimental setup details such as which optimizer was used, what data augmentation was applied, and how datasets were split. This makes the main paper harder to evaluate as a standalone document.

The relationship between ConOrd and the center loss is somewhat unclear in the main text. The center loss (Equation 8) is introduced briefly in Section 3.3 and simply added to the total loss without much discussion of why it is needed or how it interacts with the ConOrd loss. Table 5 shows that the center loss alone performs poorly and that ConOrd alone is already strong, with the combination yielding only modest improvements. This raises the question of whether the center loss is a necessary component of the framework or merely an incremental add-on, and the paper would benefit from a clearer discussion of its role.


The paper does not address ordinal regression tasks with highly imbalanced label distributions in a systematic way. The limitations section (Appendix E) acknowledges that extreme MOS values are underrepresented and cause more prediction errors, but no mitigation strategy is proposed or evaluated. Given that label imbalance is a common and practical challenge in ordinal regression, and that recent work such as SLACE (Nachmani et al., 2025) explicitly targets this issue, the lack of engagement with this problem limits the practical significance of the framework for real world datasets with skewed distributions.

While the soft weighting scheme is a meaningful contribution, the overall framework follows a well-established template: encode instances into an embedding space, apply a contrastive-style loss, regularize with a center loss, and perform k-NN inference. Each of these components is borrowed from prior work (contrastive learning from Khosla et al., supervised ordinal embeddings from Lee et al., center loss from Nguyen et al., k-NN inference from Lee et al. 2022). The primary novelty is confined to the specific formulation of the affinity and disparity weights within the contrastive loss, which, while effective, represents an incremental rather than fundamental advance over existing ordinal contrastive methods like RnC.

The paper positions ConOrd as a "general framework" for ordinal regression, but all experiments use the same k-NN inference paradigm with ℓ2-normalized embeddings on the unit sphere. It is unclear how well the proposed loss would integrate with other prediction heads (e.g., linear regression layers, ordinal classification heads) or other training paradigms (e.g., end-to-end regression with MSE loss as an auxiliary objective). Demonstrating compatibility with diverse inference strategies beyond k-NN would better support the claim of generality.

---

> ### Author Rebuttal · Authors · 2026-03-29
>
> We sincerely appreciate the reviewer’s careful reading and constructive feedback. We address each point below.
> ***
> > **Label imbalance**
>
> - **Benchmarks already contain label imbalance:**  AgeDB exhibits extreme bin imbalance (353 to 1 images per bin) and is widely used in imbalanced regression literature (Yang et al., 2021). Despite this severe skew, ConOrd achieves a state-of-the-art MAE of 5.15.
>
> - **Performance comparison to SLACE:**  We directly compare ConOrd with SLACE, a recent method designed for imbalanced ordinal regression. As shown below, ConOrd achieves better or comparable results across all datasets.
>
> |Method|MORPH II|CLAP2015|AgeDB|UTK|CACD|Adience|
> |-|-|-|-|-|-|-|
> ||MAE($\downarrow$)|MAE($\downarrow$)|MAE($\downarrow$)|MAE($\downarrow$)|MAE($\downarrow$)|MAE($\downarrow$)|
> |SLACE|2.18|2.49|5.25|3.98|4.23|**0.26**|
> |ConOrd|**1.96**|**2.46**|**5.15**|**3.92**|**4.18**|**0.26**|
>
> - **Label frequency-aware weighting:**  Incorporating label frequency can further improve performance in highly imbalanced settings. Using frequency-aware reweighting reduces AgeDB MAE from 5.15 to 5.06, as shown below. This extension, however, requires label distribution statistics, whereas ConOrd already performs strongly without such dataset-specific assumptions.
>
> |Method|$a_{ij}$|$b_{ij}$|AgeDB|
> |-|-|-|-|
> ||||MAE($\downarrow$)|
> |Proposed|$\frac{1}{(r_i - r_j)^2+\epsilon}$|$(r_i-r_j)^2$|5.15|
> |Frequency-aware|$\frac{1}{(r_i - r_j)^2+\epsilon} \cdot \frac{1}{\sqrt{f_{r_i} f_{r_j}}}$|$(r_i-r_j)^2 \cdot \sqrt{f_{r_i} f_{r_j}}$|5.06|
>
>
>
> &nbsp;
>
> > **Beyond k-NN inference**
>
> - **Compatibility with prediction heads:**
> Since ConOrd learns a metric-structured ordinal embedding space, k-NN serves as a natural nonparametric readout rather than a disconnected inference rule. However, ConOrd is not restricted to k-NN inference and also performs well with classification or regression heads, as shown below.
>
> |Method|CLAP2015|BID|BID|LSVQ-test| LSVQ-test|
> |-|-|-|-|-|-|
> ||MAE($\downarrow$)|SRCC($\uparrow$)|PCC($\uparrow$)|SRCC($\uparrow$)|PCC($\uparrow$)|
> |$L_{\text{CE}}$|2.605|0.742|0.762|0.902|0.901|
> |$L_{\text{CE}}+L_{\text{ConOrd}}+L_{\text{center}}$|2.599|0.876|0.881|0.904|0.903|
> |$L_{\text{MAE}}$|2.506|0.881|0.900|0.903|0.902|
> |$L_{\text{MAE}}+L_{\text{ConOrd}}+L_{\text{center}}$|2.481|0.899|0.914|0.904|0.904|
>
> - **Effectiveness as an auxiliary loss:**  We further verify ConOrd’s plug-and-play property on LoDa without architectural modifications.
>
> |CLIVE dataset|SRCC($\uparrow$)|PCC($\uparrow$)|
> |-|-|-|
> |$L_{\text{plcc}}$ (original LoDa)|0.876|0.896|
> |**$L_{\text{plcc}}+L_{\text{ConOrd}} $**| **0.880** |**0.900**|
>
> &nbsp;
>
> > **Weighting design**
>
> We agree that the quadratic form is not the only possible choice. However, it provides a simple instantiation that satisfies a key monotonic transition condition: as the rank gap increases, the interaction shifts consistently from attraction to repulsion. The comparable performance of alternatives in Table 6 therefore suggests that the benefit comes from this continuous distance-aware mechanism itself, rather than from a specific formula. We also provide a more detailed gradient-based analysis of this point in our response to Reviewer FSjq and will incorporate it into the revised manuscript.
>
> &nbsp;
>
> > **Center loss**
>
> $L_{\text{ConOrd}}$ models inter-rank relationships, while $L_{\text{center}}$ encourages intra-rank compactness. Since $L_{\text{ConOrd}}$ primarily constrains relative ordering across ranks, it does not explicitly enforce tight clustering within each rank. Thus, $L_{\text{center}}$ is an auxiliary rather than essential regularizer: it reduces intra-rank variance and provides a modest but consistent gain.
>
> &nbsp;
>
> > **Novelty of the overall framework**
>
> We agree that ConOrd is built from familiar ingredients such as embedding-based learning, contrastive objectives, regularization, and k-NN inference. Our novelty claim is therefore not that each component is individually new, but that the proposed objective introduces a different formulation for incorporating ordinal structure into contrastive learning: continuous, distance-aware, all-pair supervision, instead of threshold-based or partially active ordinal supervision. This formulation allows ConOrd to exploit the full ordinal spectrum within a batch, rather than a subset of selected pairs. We will clarify this formulation-level distinction more explicitly in the revised manuscript.
>
> &nbsp;
>
> > **Subgroup analysis**
>
> We conduct subgroup analysis on the AgeDB test set to examine potential demographic bias and will include these results in the revision.
>
> |Gender group|# samples|MAE($\downarrow$)|
> |-|-|-|
> |Female|867|5.505|
> |Male|1273|4.900|
>
> |Age group|# samples|MAE($\downarrow$)|
> |-|-|-|
> |0–19|158|5.70|
> |20–39|600|4.50|
> |40–59|600|5.09|
> |60–79|595|5.19|
> |80+|187|6.77|
>
>
>
> ***
> We would be happy to clarify any remaining points if needed. Thank you again for your feedback.

---

> > ### Author Rebuttal · Reviewer_ogaZ · 2026-04-04
> >
> > While the paper presents a contrastive learning framework for ordinal regression, I recommend a weak rejection due to limited theoretical novelty and significant flaws in the empirical evaluation. The mathematical and technical contributions offer limited novelty beyond existing distance-aware and contrastive ordinal regression methods. More critically, the evaluation lacks fairness because the proposed method is compared against standard regression approaches using mismatched backbones. To properly interpret the added value of the new objective function rather than the representational power of the backbone itself, comparisons must be conducted under strictly matched architectures and identical training protocols. Finally, the specific choice of evaluation datasets does not adequately support the authors' claim of proposing a "general" ordinal regression framework, leaving the comprehensive evaluation lacking and the fundamental efficacy of the method unproven.

---

> > > ### Author Response · Authors · 2026-04-04
> > >
> > > We sincerely thank the reviewer for the continued engagement and for raising these new points regarding "backbone fairness" and "dataset selection." As these concerns were the primary focus of Reviewer **AXys**, we believe we have already addressed them successfully in our rebuttal to them. For your convenience, we repeat the relevant clarifications and empirical results below.
> > >
> > > ***
> > >
> > > > **Backbone fairness**
> > >
> > > We would like to clarify that the comparisons in Table 4 were already conducted under a fully controlled setting, where:
> > > - all methods share the *same backbone (ViT-B)*,
> > > - *identical training protocols* are used across all baselines, and
> > > - a *unified k-NN inference scheme* is adopted for evaluation.
> > >
> > > This design ensures that the performance differences are attributable solely to the loss formulation, rather than architectural or training discrepancies.
> > >
> > > |Method|CLAP2015|BID|BID|LSVQ-1080p|LSVQ-1080p|
> > > |-|-|-|-|-|-|
> > > ||MAE($\downarrow$)|SRCC($\uparrow$)|PCC($\uparrow$)|SRCC($\uparrow$)|PCC($\uparrow$)|
> > > |$L_{\text{SupCon}}$|2.625|0.819|0.876|0.614|0.682|
> > > |$L_{\text{OCL}}$|2.597|0.909|0.921|0.697|0.687|
> > > |$L_{\text{MMNP}}+L_{\text{CE}}$|2.777|0.828|0.825|0.716|0.727|
> > > |$L_{\text{RnC}}$|2.531|0.892|0.906|0.812|0.843|
> > > |$L_{\text{ConOrd}}+L_{\text{center}}$(Proposed)|**2.461**|**0.913**|**0.925**|**0.818**|**0.851**|
> > >
> > > &nbsp;
> > >
> > > > **Regarding the choice of evaluation benchmarks**
> > >
> > > We respectfully disagree with the concern that our benchmark selection does not support the claim of a general ordinal regression framework.
> > >
> > > - **Age estimation** is one of the most established benchmarks in ordinal and imbalanced regression literature.
> > > - **IQA/VQA** are also standard ordinal regression tasks with continuous, ordered labels. Our goal in including these benchmarks is not to claim task-specific architectural novelty, but to examine whether the proposed ordinal contrastive loss remains beneficial in realistic ordinal regression settings. Rather than weakening the generality claim, these experiments broaden the empirical evidence for it.
> > >
> > > &nbsp;
> > >
> > > > **Comparison to DIR methods under matched settings (Updated)**
> > >
> > > To directly address your concern regarding fairness, we conducted additional controlled experiments on standard deep imbalanced regression (DIR) benchmarks.  All methods are implemented using the same ResNet-50 backbone for AgeDB-DIR and IMDB-WIKI-DIR:
> > >
> > > **Updates:** We further include STS-B-DIR and NYUD2-DIR results. Following prior work, we use a BiLSTM with GloVe embeddings for STS-B-DIR and a ResNet-50-based encoder–decoder [1] for NYUD2-DIR.
> > >
> > > [1] Hu et al. "Revisiting single image depth estimation: Toward higher resolution maps with accurate object boundaries," WACV 2019.
> > >
> > > |Method|AgeDB-DIR|IMDB-WIKI-DIR|STS-B-DIR|NYUD2-DIR|
> > > |-|-|-|-|-|
> > > ||MAE($\downarrow$)|MAE($\downarrow$)|MSE($\downarrow$)|RMSE($\downarrow$)
> > > |RankSim|7.02|7.50|0.903|-|
> > > |Ordinal Entropy|7.46|-|-|-|
> > > |Rank-n-Contrast|6.14|-|-|-|
> > > |ConR|7.20|7.33|-|1.304|
> > > |VIR|6.99|7.19|0.892|1.305|
> > > |SRL|7.22|7.69|0.877|-|
> > > |ConOrd (Proposed)|**6.02**|**7.06**|**0.868**|**1.294**|
> > >
> > > In addition, we performed comparisons on IQA/VQA benchmarks under the same backbone and evaluation pipeline as ConOrd:
> > >
> > > |Method|BID|BID|LSVQ-1080p|LSVQ-1080p|
> > > |-|-|-|-|-|
> > > ||SRCC($\uparrow$)|PCC($\uparrow$)|SRCC($\uparrow$)|PCC($\uparrow$)|
> > > |RankSim|0.892|0.909|0.814|0.848|
> > > |Ordinal Entropy|0.886|0.909|0.809|0.845|
> > > |Rank-n-Contrast|0.892|0.906|0.812|0.843|
> > > |ConR|0.885|0.909|0.812|0.846|
> > > |SRL|0.900|0.921|0.814|0.847|
> > > |ConOrd (Proposed)|**0.913**|**0.925**|**0.818**|**0.851**|
> > >
> > > Across all settings, ConOrd consistently outperforms competing ordinal/DIR methods under identical conditions, confirming that the observed gains stem from the proposed objective.
> > >
> > > We will include these additional results and explicitly describe the unified evaluation protocol in the revised manuscript.
> > >
> > >
> > > ***
> > > Thank you for your engagement throughout the review process. We hope the strictly controlled empirical evidence and clarifications provided above fully resolve your remaining concerns.

---

### Decision · Program_Chairs · 2026-04-30

**Decision:**

Accept (regular)

**Comment:**

The paper proposes a method for ordinal regression with a novel contrastive learning framework. Reviewers generally feel the method is incremental, but it still achieves good results. The most critical concerns from the reviewers were the validity and coverage of the evaluation. The authors provided updated results and clarifications, and this updated content should be clearly incorporated into the final version if the paper is accepted. The paper is recommended for acceptance at ICML as a weak accept.